# Therapeutic Potential of Prenylated Flavonoids of the Fabaceae Family in Medicinal Chemistry: An Updated Review

**DOI:** 10.3390/ijms252313036

**Published:** 2024-12-04

**Authors:** Jaime Morante-Carriel, Hugo Nájera, Antonio Samper-Herrero, Suzana Živković, María José Martínez-Esteso, Ascensión Martínez-Márquez, Susana Sellés-Marchart, Anna Obrebska, Roque Bru-Martínez

**Affiliations:** 1Plant Proteomics and Functional Genomics Group, Department of Biochemistry and Molecular Biology and Soil and Agricultural Chemistry, Faculty of Science, University of Alicante, Carretera San Vicente del Raspeig s/n, 03690 San Vicente del Raspeig, Alicante, Spain; hnajera@cua.uam.mx (H.N.); antonio.samper@ua.es (A.S.-H.); mjose.martinez@ua.es (M.J.M.-E.); asun.martinez@ua.es (A.M.-M.); susana.selles@ua.es (S.S.-M.); anna.obrebska@ua.es (A.O.); roque.bru@ua.es (R.B.-M.); 2Plant Biotechnology Group, Faculty of Forestry and Agricultural Sciences, Quevedo State Technical University, Av. Quito km. 1 1/2 vía a Santo Domingo de los Tsáchilas, Quevedo 120501, Ecuador; 3Departamento de Ciencias Naturales, Universidad Autónoma Metropolitana—Cuajimalpa, Av. Vasco de Quiroga 4871, Colonia Santa Fe Cuajimalpa, Alcaldía Cuajimalpa de Morelos, Ciudad de México 05348, Mexico; 4Alicante Institute for Health and Biomedical Research (ISABIAL), 03010 Alicante, Alicante, Spain; 5Institute for Biological Research “Siniša Stanković”-National Institute of Republic of Serbia, University of Belgrade, Bulevar Despota Stefana 142, 11108 Belgrade, Serbia; suzy@ibiss.bg.ac.rs; 6Research Technical Facility, Proteomics and Genomics Division, University of Alicante, 03690 San Vicente del Raspeig, Alicante, Spain; 7Multidisciplinary Institute for the Study of the Environment (IMEM), University of Alicante, 03690 San Vicente del Raspeig, Alicante, Spain

**Keywords:** prenytaled flavonoids, natural compounds, bioactivity, Fabaceae, medicinal plants

## Abstract

Much attention has been paid to the potential biological activities of prenylated flavonoids (PFs) in various plant families over the last decade. They have enormous potential for biological activities, such as anti-cancer, anti-diabetic, antimicrobial, anti-inflammatory, anti-Alzheimer’s, and neuroprotective activities. Medicinal chemists have recently shown a strong interest in PFs, as they are critical to the development of new medicines. PFs have been rapidly prepared by isolation and semi- or full synthesis, demonstrating their significant utility in medicinal chemistry research. This study encompasses the research progress on PFs in the last decade, including their pharmacological activities in the Fabaceae family. This information demonstrates the bioactive potential of PF compounds and their role in the control and treatment of various human health problems.

## 1. Introduction

Medicinal plants, widely used worldwide, have strong bioactive properties. Flavonoids and prenylated compounds in these plants are responsible for some health-promoting properties [1,2,3,4,5]. Flavonoids are phenolic compounds with a basic flavane structure of 15 carbon atoms (C6-C3-C6) consisting of two benzene rings (A and B) linked by a three-carbon pyran ring (C) (Figure 1a). The position of the catechol B-ring on the C-pyran ring, and the number and arrangement of hydroxy groups in the catechol group of the B-ring influence the antioxidant capacity of the flavonoids. They have strong antioxidant properties and other beneficial effects, including anti-cancer, anti-inflammatory, anti-obesity, and cardiovascular disease prevention [6].

Prenylated flavonoids (PFs) belong to a unique category of prenylated phenolic compounds. They contain at least one prenyl group attached to the flavonoid structure and exhibit various biological effects. In most cases, the prenyl group enhances the biological properties of the flavonoids (Figure 1b). Consequently, the potential of PFs for development and application is very high. 

The Fabaceae family, formerly known as Leguminosae, is the third largest family of flowering plants, with over 700 genera and almost 20,000 species worldwide [7]. These plants are distributed in the Mediterranean, subtropical savannahs, and tropical and dry forests [8,9]. Research on the phytochemical content and antioxidant properties of the Fabaceae family has mainly focused on crop plants (e.g., *Glycine max*, *Medicago sativa*, *Vicia faba*, etc.), which are important food sources for humans [10,11,12]. Fabaceae plants are frequently used in traditional medicine (e.g., *Glycyrrhiza uralensis*, *Desmodium caudatum*, *Millettia extensa*, *Dalea searlsiae*, and *Dalea elegans*, among others). The phytochemicals present in this family, such as PFs, contribute significantly to their biological activities. 

The majority of prenylated phenolic compounds have been recognized as flavonoids, stilbenoids, and xanthones. C-prenylated derivatives are more common than *O*-prenylated derivatives [13]. However, a thorough analysis of the scientific literature reveals that there are still many unanswered questions about these fascinating bioactive compounds and new targets for their beneficial effects are constantly being discovered. Although several reviews on prenylated phenolics can be found in the scientific literature [13,14,15], work still needs to focus on the intriguing bioactive properties of prenylated phenolic compounds from Fabaceae plants.

This review summarized 260 PFs obtained from several species of the Fabaceae family over the past decade. The PFs were classified into five bioactivities: anti-cancer, antimicrobial, anti-inflammatory, anti-diabetic, and anti-Alzheimer’s and neuroprotective activities. In addition, 11 PF compounds showed more than one biological activity, as indicated in Table 1. 

This review aims to present the latest scientific findings on the biological activities of PF compounds of Fabaceae plants and the associated health benefits supported by experimental evidence from the last decade.

## 2. Biosynthetic Pathway of Prenylated Flavonoids 

Flavonoids play an important role in plant development and defense, mainly synthesizing specialized metabolites known as phytoalexins. When pathogens or other stressors attack the plant, these compounds, derived from L-phenylalanine and L-tyrosine, produce a series of intricate enzymatic reactions, each catalyzed by a specific enzyme. In this context, prenylated flavonoids are a special type of flavonoid derivative that is characteristic of modification by prenylation of the skeleton. Prenylation refers to alkyl-substituted groups such as prenyl, geranyl, lavandulyl, and farnesyl groups, which have a greater potential for further modifications such as oxidation, cyclization, and hydroxylation, and increase the structural and biological diversity of PFs. The biosynthesis of PFs is a series of biochemical reactions catalyzed by specific enzymes: phenylalanine ammonia-lyase, tyrosine ammonia-lyase, cinnamate 4-hydroxylase, cytochrome P450 reductase, and 4-coumarate-CoA ligase. These enzymes convert the initial amino acid into coumaroyl-CoA, a precursor for various PFs [16]. A detailed schema of the metabolic pathway of PFs in plants is shown in Figure 2.

## 3. Biological Activities of Prenylated Flavonoids

PFs are a unique class of flavonoids that exist specifically as the self-defense strategy of plants [17,18]. They also have a rather lipophilic structure, which leads to their high affinity for cell membranes and enhancement of biological activity [17]. This review describes the biological activities of PFs in Fabaceae plants, including anti-cancer, anti-diabetic, anti-inflammatory, antimicrobial, and anti-Alzheimer’s activities, as well as their neuroprotective activities. We conducted a systematic literature search in English for a 10 year timespan (2014–2023) using PubMed and Web of Knowledge. The search terms used were “Fabaceae prenylated flavonoids”, “Fabaceae prenyl flavonoids”, and “Fabaceae prenylation flavonoids”, as well as their activity combinations, throughout multiple search engines. Most of the articles collected in these two databases focused on the PF compounds of the Fabaceae family (Figure 3a). All articles with little experimental evidence were excluded. Several PF compounds were identified by Fabaceae species and biological activities (Figure 3b).

### 3.1. Anti-Cancer Activity

Cancer, a leading cause of death and a significant health burden globally, has prompted high treatment costs for affected individuals [19]. The International Agency for Research on Cancer predicted the global prevalence of 36 types of cancers to reach 29.8 million by 2020, with 19.3 million new cancer cases and 10 million cancer deaths [20]. The characteristics of cancer, including apoptosis [21], angiogenesis, numerous replications [22,23], synthesis of growth signals [24], resistance to growth inhibitory signals, and metastasis, enable cancer cells to proliferate continuously, survive over a long period, and displace normal cells. However, the potential of natural compounds as alternative anti-cancer agents is a promising avenue for research. Numerous studies have confirmed the anti-cancer effects of these natural bioactive chemicals [25,26]. In this context, new PFs with different biological activities against different types of cancer have been recognized as compounds with potential cytotoxicity against various cancer cells.

Several PFs (sinopodophyllines B; nitidulin; maakiaflavanones A-B; sophoraflavanone G; alopecurones J, P; renifchalcone A; renifolins D-H; mappianthone A; robustone methyl ether; dihydrorobustone methyl ether; indicanine E; pumilaisoflavone D; 5-*O*-methyl-2′-methoxy-3′-methyl alpinumisoflavone; erythraddison A) from *Sophora flavescens*, *Dalbergia velutina*, *Maackia amurensis*, *Sophora pachycarpa*, *Desmodium renifolium*, and *Mappianthus iodoides* have been identified [15,27,28,29,30,31,32]. These compounds were evaluated for their cytotoxicity, a measure of their ability to kill or inhibit the growth of cancer cells, against several human cancer cell lines in vivo and in vitro [15,27,28,29,30,31,32]. Some structures of prenylated flavonoids from the Fabaceae family with anti-cancer activity are shown in Figure 4.

While most compounds showed relatively low activity at the half-maximal inhibitory concentration (IC_50_), 5-*O*-methyl-2′-methoxy-3′-methylalpinumisoflavone stood out with a remarkable value of 0.98 μM against HL-60 cancer cells [33]. The unique structures of these compounds, including the alpinumisoflavone in the B-ring, significantly contribute to their cytotoxicity against cancer cells [34].

Licoflavonol compounds extracted from the medicinal plant *G. uralensis* showed cytotoxicity against SW480 cells with IC_50_ values of 14.53 µM [35]. Additionally, 6,8-diprenyleriodictyol, a natural compound isolated from the plant *Derris ferruginea*, showed cytotoxicity against MRC-5 and KB cells, with IC_50_ values of 8.0 and 8.5 µM, respectively [36]. Lonchocarpol A and dorsmanine I, also obtained from the stems of *D. ferruginea*, inhibited the growth of MRC-5 and KB cells, with IC_50_ values between 6.2 and 23.8 µM.

Two compounds, maackiafavanone and 5-hydroxysophoranone, were found to have a toxic effect on SK-MEL-5 and HeLa cells, with IC_50_ values of 7.7 and 12 µM, respectively. Similar compounds, such as isomaackiafavanone A, isomaackiafavanone B, and abyssinone V, displayed lower cytotoxicity towards these cells with IC_50_ values in the 16–36 µM range. This finding suggests that an 8-prenyl side chain may increase cytotoxicity, while a methoxy group at C-2′ of the B-ring showed the opposite effect [29].

Three diprenylated flavonoids, addisoniafavanones I, II, and 5,7-dihydroxy-5′-prenyl-[2″,2″-(3″-hydroxy)-dimethylpyrano]-(5″,6″:3′,4′)flavanone, have been extracted from *Erythrina addisoniae*. These compounds exhibited stronger toxicity to H4IIE hepatoma cells than monoprenyl-substituted flavonoids, reaching EC_50_ values of 14.7, 5.25, and 8.5 µM, respectively. Both addisoniafavanones triggered apoptosis by activation of caspase-3/7 [37].

Two closely related chalcones, sanjuanolide and sanjoseolide, were extracted from *Dalea frutescens* and showed different cytotoxicity levels against prostate cancer cells PC-3 and DU 145. Compared to sanjoseolide, cannabisanolide has a 3.2- and 3.6-fold more substantial effect on PC-3 and DU 145 cells, respectively. The main difference between these compounds is that the 2-hydroxy-3-methyl-3-butanyl group in sanjuanolide was replaced by 2,3-dihydroxy-3-methyl-3-butenyl group in sanjoseolide. This result implied that hydroxylation of the prenyl side chain reduces cytotoxicity [38]. In addition, a prenylated chalcone from the seeds of *Millettia pachycarpa* named 3-hydroxy-4-methoxylonchocarpin showed cytotoxicity against K562 cells with an IC_50_ value of 2.4 µg mL^−1^ [39].

Chalcone compounds (artonin ZA-1, artonin ZA-2, renifolin C, 2′,4-hydroxy-3,4′-dimethoxychalcone, 2′,3-hydroxy-4,4′-dimethoxychalcone, and renifolins D–H) obtained from *D. renifolium* and *Desmodium podocarpum* exhibited significant cytotoxic effects against NB4, A549, SHSY5Y, PC3, and MCF-7 cells [32,40,41]. In addition, the compounds kanzonol C and hedysarumine F from *Hedysarum gmelinii* showed remarkable cytotoxicity against A549 cells, with IC_50_ values of 9.67 and 7.79 μM, respectively. Kanzonol C also exhibited strong cytotoxic activity against HCT116 cells, with an IC_50_ value of 8.85 μM [42].

Flemiphilippinone C, which contains two prenyl groups at C-3′, was extracted from the roots of *Flemingia philippinensis*, and it had been shown to have antiproliferative activity against PC-3, Bel-7402, and CaEs-17 cancer cells, with GI_50_ values of 14.12, 1.91, and 2.58 μM, respectively. Flemiphilippinone C induces apoptosis in Bel-7402 cells by increasing S/G2 arrest via a specific mitochondria-related pathway [43]. This mechanism of action provides valuable insight into the potential of flemiphilippinone C as an anti-cancer agent. Gancaonin G isolated from *G. uralensis* showed cytotoxicity against SW480 cells, with an IC_50_ value of 9.84 μM. In addition, 4′-hydroxy-5,7-dimethoxy-6-(3-methyl-2-butenyl)-isoflavone, gancaonin N, isopiscerythrone, viridiflorin, and ficucaricone D demonstrated cytotoxicity against five human cancer cell lines, namely HL-60, SMMC-7721, A-549, MCF-7, and SW480, with IC_50_ values ranging from 0.42 μM to 8.48 μM [44].

Auriculasin extracted from the roots of *F. philippinensis* had been shown to have an inhibitory effect on PC-3 cells, with a GI_50_ value of 8.33 μM. On the other hand, a chromone-derivative, eriosematin, did not show any activity (GI_50_ > 100 μM). According to the latter findings, auriculasin B-ring with a 3′,4′-dihydroxy group could be a key component in preventing the development of PC-3 cells [43]. MCF-7, A549, and HepG2 cells were exposed to the cytotoxicity of flemiphilippinin G and 5,7,3′-trihydroxy-2′-(3-methylbut-2-enyl)-4′,5′-(3,3-dimethylpyrano) isoflavone, with IC_50_ values ranging from 4.8 to 24.8 μM [36]. With IC_50_ values ranging from 0.1 μM to 9.0 μM, tephrosin, cis-(6aβ,12aβ)-hydroxyrotenone and rotenone isolated from *Indigofera spicata* showed cytotoxicity against HT-29, 697 humans acute lymphoblastic leukemia, and Raji human Burkitt’s lymphoma cells. Furthermore, neither rotenone nor cis-(6aβ,12aβ)-hydroxyrotenone (IC_50_ of 0.1 μM against HT-29 cells) significantly affected the viability of normal CCD-112CoN cells (IC_50_ > 100 μM) [45].

The PF compound glyurallin A, derived from G. uralensis, demonstrated cytotoxicity against SW480 cells, with an IC_50_ value of 10.86 μM [35]. Five human cancer cell lines (SMMC-7721, SW480, HL-60, A-549, and MCF-7) were subjected to substantial cytotoxic effects by asthonningine A, indicanine B, ficucaricones A and B, 3″,4″-dihydrothonningine C, and indicanine B, with IC_50_ values ranging from 0.18 μM to 18.76 μM [44]. Finally, alopecurone J showed a cytotoxic effect against HeLa, HCT116, A2780, and A549 cells, with IC_50_ values ranging from 9.97 to 30.91 μM [46].

Compounds derived from different plants, such as *G. uralensis*, *D. ferruginea*, *E. addisoniae*, *D. frutescens*, and others, exhibited varying levels of cytotoxicity against different human cancer cell lines. This indicates the potential diversity of natural compounds as sources of anti-cancer agents. Notably, two PFs (glabrene and glabridin) were isolated from *G. glabra*, and the estrogen-like activities of both compounds were tested by three systems including competitive ligand-binding assays, in vitro cell assays, and in vivo animal models [47]. The outcome indicated that these compounds closely bind with the estrogen receptor, with EC50 values of 5 × 10^−5^ M and 5 × 10^−6^ M, respectively. Additionally, Aregueta-Robles et al. (2018) found that *Phaseolus vulgaris* extract and its flavonoid contents have an inhibitory effect on lymphoma in mice both in vivo and in vitro [48]. Ombra et al. (2016) also confirmed that flavonoids from *P. vulgaris* showed considerable anti-cancer properties and suppressed the development of human MCF-7, while flavonoids also showed an inhibitory effect against human epithelial colorectal adenocarcinoma (Caco-3) cells during in vivo model studies. Also, various forms of lectins showed anti-cancer properties under in vivo conditions [49].

Some compounds, such as flemiphilippinone C from *F. philippinensis* and gancaonin G from *G. uralensis*, showed promising antiproliferative and cytotoxic effects against certain cancer cell lines, indicating their potential as candidates for further investigation as anti-cancer agents. 

In summary, the research highlights the diverse sources of natural compounds with potential anti-cancer properties and the critical importance of understanding the structure–activity relationships of these compounds for the development of effective cancer treatments. Further studies and clinical trials are essential to harness these natural compounds for cancer treatment. These and other PF compounds with anti-cancer activity of the Fabaceae family are listed in Table 2.

### 3.2. Antimicrobial Activity

With the escalating issue of bacterial antibiotic resistance due to the overuse of antibiotics, the potential of plants from the Fabaceae family, renowned for their antimicrobial properties, offers a ray of hope. These plants are invaluable resources for identifying potential antibacterial agents. The molecular diversity of phytochemicals from Fabaceae extracts and isolated compounds, enhanced by germination and fungal attacks, presents a promising avenue for combating antibiotic resistance. The relationship between phytochemical composition and the antibacterial properties of extracts and isolated compounds has been extensively investigated, with extracts rich in prenylated isoflavonoids and stilbenoids showing potent antibacterial activity against various bacteria and/or fungi [52,53,54].

The compounds known as iconisoflavan and iconisoflaven, as well as four other PFs, have been found in the roots of *Glycyrrhiza iconica*; using a microdilution method, these compounds were tested for their antimicrobial properties against five bacteria and one yeast line. Of these, iconisoflavan, (3*S*)-licoricidin, licorisoflavan A, and topazolin displayed significant activity against *Salmonella typhimurium* ATCC 13311, with minimum inhibitory concentrations (MIC, a measure of the effectiveness of an antimicrobial in inhibiting the growth of a microorganism) ranging from 2 to 8 μg mL^−1^ [55].

Several PFs were extracted from the roots, stem bark, and leaves of *Eriosema montanum*. The antibacterial activity of the crude extracts and isolated constituents was determined using a comprehensive method that tests against both Gram-positive and Gram-negative bacteria. Of the compounds tested, two new prenylated dihydrochalcones (2′,4′,5,6′-tetrahydroxy-4-methoxy-3,3′-diprenyldihydrochalcone,2′,4′,4,6′-tetrahydroxy-3,3′-diprenyldihydrochalcone and lupinalbin A) exhibited robust activity against the Gram-positive bacteria *Bacillus subtilis,* with MIC values ranging from 3.1 μM to 8.9 μM, as did the crude leaf extract (MIC of 3.0 μg mL^−1^). However, neither crude extracts nor isolated compounds show activity against *Escherichia coli* [56].

The escalating resistance to conventional antibiotics necessitates a shift towards new, natural treatment methods against methicillin-resistant *Staphylococcus aureus* (MRSA). In vitro studies have delved into the anti-MRSA properties of 23 PFs from the Fabaceae family. The results are significant, revealing the most effective compounds against MRSA including the di-prenylated flavonoids glabrol and 6,8-diprenylgenistein, and the mono-prenylated 4′-*O*-methyl glabridin, all with MIC values ≤ 10 μg mL^−1^ (30 μM). The structures of PFs from the Fabaceae family with high activity against MRSA are shown in Figure 5, providing a comprehensive understanding of their potential in combating MRSA.

This research has uncovered the importance of formal charge, hydrophobic volume, and hydrogen bonding in anti-MRSA activity, and suggested different modes of action of different PFs. These findings could potentially revolutionize the field of microbiology, pharmacology, and drug development, inspiring the development of new anti-MRSA therapeutics. The potential impact of this research is significant, paving the way for developing new, effective treatments against MRSA [57].

In another study, seven isoflavone derivatives were extracted from *Erythrina lysistemon*. These derivatives, erybraedin A, phaseollidin, abyssinone V-4′-methyl ether, eryzerin C, alpumisoflavone, cristacarpin, and lysisteisoflavone, were tested against various skin pathogenic bacterial species.

The most important finding about the PF compounds tested was the wide range of MIC values found, which ranged from 1 to 600 μg mL^−1^. However, the results showed no specific selectivity pattern for different Gram-types [58].

Six PFs, specifically malheurans A−D, (2*S*)-5′-(2-methylbut-3-en-2-yl)-8-(3-methylbut-2-en-1-yl)-5,7,2′,4′-tetrahydroxyflavanone and prostratol F, have been isolated from *D. searlsiae*. Antimicrobial assays have shown that these compounds are effective against *Streptococcus mutans*, *Bacillus cereus*, and oxacillin-sensitive (OSSA) and -resistant (ORSA) *S. aureus*, with MICs of 2 to 8 μg mL^−1^ [59].

*Vatairea guianenis*, a medicinal plant from the Amazon region, is traditionally used to treat skin diseases. A leaf extract of *V. guianensis* and its sub-extracts exhibited remarkable antibacterial and antifungal properties. Çiçek et al. (2022) discovered one known and three new isoflavones, all prenylated at position C-8, one of which displayed significant activity (IC_50_ value ranging from 6.8 μM to 26.9 μM). The results of this study support the traditional use of *V. guianensis* for treating skin diseases [54].

The 5-deoxy-3′-prenylbiochanin A, erysubin F, and 7,4′-dihydroxy-8,3′-diprenylflavone were tested against some bacterial strains and one fungal pathogen. None of these three compounds was found to have antimicrobial activity against *Salmonella enterica*, *E. coli*, or *Candida albicans* (MIC values above 80.0 μM). However, the diprenylated natural product erysubin F and its flavone isomer 7,4′-dihydroxy-8,3′-diprenylflavone demonstrated notable in vitro activity against MRSA with MIC values of 15.4 μM and 20.5 μM, respectively. At the same time, monoprenylated compound 5-deoxy-3′-prenylbiochanin A showed no activity against MRSA [60].

The PF compounds (E)-5-hydroxytephrostachin, terpurlepflavone, and tachrosin, extracted from the stem of the commonly used medicinal plant *Tephrosia purpurea* subsp. *leptostachya,* possessed antiplasmodial activity against the chloroquine-sensitive D6 strains of *P. falciparum*. Of these, (*E*)-5-hydroxytephrostachin exhibited the highest activity with an IC_50_ of 1.7 ± 0.1 μM [61]. Five new compounds, namely rhodimer, rhodiflavan A, rhodiflavan B, rhodiflavan C, and rhodacarpin, were isolated from the roots of *Tephrosia rhodesica*. Among these compounds, rhodimer was an atypical flavanone-flavan dimer, while rhodiflavan C possessed a five-membered A-ring, a unique feature for this genus. The crude extract from the root and some of its isolated components exhibited antimicrobial activity against the chloroquine-sensitive (3D7) strain of *Plasmodium falciparum* [62].

Various compounds were extracted from *D. caudatum*, including seven new PFs and nineteen known compounds, and subsequently tested for their ability to inhibit the growth of film-forming yeast *Zygosaccharomyces rouxii* F51. Some of the tested compounds exhibited inhibitory effects, and their MIC values ranged from 7.8 to 62.5 μg mL^−1^. Of them, 2″,2″-dimethylpyran-(5″,6″:7,8)-5,2′-dihydroxy-4′-methoxy-(2*R*,3*R*)-dihydroflavonol exhibited pronounced inhibitory activity, with an MIC value of 7.8 μg mL^−1^ [63].

The plant *Glycyrrhiza inflata* produces two PF compounds, 2-(3-methyl-2-butenyl)-3,5,4′-trihydroxy-bibenzyl and (2*S*)-6-[(*E*)-3-hydroxymethyl-2-butenyl]-3′,4′,5,7-tetrahydroxy-dihydroflavanone, as well as some dihydroflavanones, which have been shown to inhibit the growth of both *Staphylococcus epidermidis* and *S. aureus*. The compound 2-(3-methyl-2-butenyl)-3,5,4′-trihydroxy-bibenzyl possessed moderate antibacterial activity against *S. epidermidis* (MIC of 12.50 mg mL^−1^) and *S. aureus* (MIC value 50.0 mg mL^−1^). Further analysis of their structure–activity relationships showed that the antibacterial properties of dihydroflavanones were mainly influenced by the position of the prenyl group [64].

The research emphasizes the potential of plants from the Fabaceae family as a source of valuable antibacterial agents. The compounds isolated from these plants exhibited significant antibacterial and antifungal properties against specific bacteria and fungi, especially certain compounds that displayed effective anti-MRSA properties, indicating the potential for new anti-MRSA therapeutics. The results also underline the importance of exploring natural treatments, especially in the face of increasing antibiotic resistance. Overall, this study provides valuable insights into the antimicrobial potential of phytochemicals from Fabaceae extracts and their potential impact on advancing new therapeutics in microbiology and drug development. Table 3 gives an overview of the antimicrobial active PF compounds of the Fabaceae family.

### 3.3. Anti-Inflammatory Activity

Various inflammatory mediators, including ROS (reactive oxygen species), interleukin (IL)-1β, TNF-α (tumor necrosis factor-α), IL-6, NO (nitric oxide), and NF-k, regulate the complex process of inflammation in the body [73]. Despite the plethora of pharmaceuticals that combat inflammation through various mechanisms, the effective treatment of inflammatory diseases remains a problem that requires the exploration of novel agents whose mode of action differs from classical anti-inflammatory medications. The consistent anti-inflammatory properties of PFs and their derivatives offer a promising avenue for the future of anti-inflammatory treatments [74]. In this context, several PF compounds (sophoraflavanone G, kurarinone, kuraridin, 5-methyisophoraflavanone B, echinoisoflavanone, echinoisosophoranone, isosophoranone, sanggenon B, and sanggenon D) and derivatives have been isolated from Fabaceae plants, and their anti-inflammatory potential has been evaluated [75,76].

From a mechanistic point of view, (*2R*)-3α,7,4′-trihydroxy-5-methoxy-8-(γ,γ-dimethylallyl)-flavanone (SFM) was found to simultaneously inhibit two important inflammatory signaling pathways, NF-κB and JNK/AP-1. SFM effectively inhibited the phosphorylation and degradation of IκBα by preventing the subsequent translocation of p65 and attenuating NF-κB activity. At the same time, it suppressed JNK phosphorylation, resulting in the inhibition of AP-1 transcriptional activity. These results provide a scientific basis for the traditional use of the anti-inflammatory herb *S. flavescens* Ait and emphasize the potential of SFM as a natural candidate for treating inflammatory conditions [77].

Research on the PF compound sophoraflavanone G (SG) has provided unique insights into its mechanism of action. The expression of tumor necrosis factor-α (TNF-α) and the proinflammatory cytokines interleukin-1β (IL-1β) and interleukin-6 (IL-6) was significantly reduced at the protein and gene level. In addition, SG demonstrated its potential by inhibiting the lipopolysaccharide (LPS)-induced upregulation of phosphorylated phosphoinositide-3-kinase and Akt (PI3K/Akt). The efficacy of SG was observed by attenuating the expression of phosphorylated Janus kinase signal transducer and activator of transcription (JAK/STAT). Additionally, SG played a regulatory role by upregulating the expression of heme oxygenase-1 (HO-1) via a unique mechanism of nuclear translocation of factor E2-related factor 2(Nrf2) [78].

Kushenol C (KC) was found to reduce the production of NO, PGE2, IL-6, IL1β, MCP-1, and IFN-β in LPS-stimulated RAW264.7 macrophages in a dose-dependent manner. The inhibition of STAT1 and STAT6, and the activation of NF-κB by KC were considered to cause the inhibition of IFN-β, IL1β, IL-6, NO, MCP-1, and PGE2 in LPS-stimulated RAW264.7 macrophages. KC also increased the activities of HO-1 and its expression while it upregulated the transcriptional activities of Nrf2 in LPS-stimulated RAW264.7 macrophages [79].

*Psoralea corylifolia* is frequently used in traditional medicine to treat inflammation and infectious diseases. Isobavachin, a bioavailable prenylated flavonoid from *P. corylifolia*, has numerous biological properties, but very little is known about its anti-inflammatory effects and mechanisms of action [80]. The anti-inflammatory effects of isobavachin reduced the overproduction of inflammatory mediators both in vitro and in vivo, as well as proinflammatory cytokines, inducible nitric oxide synthase (iNOS), cyclooxygenase-2 (COX-2), mitogen-activated protein kinase (MAPK) phosphorylation and nuclear translocation of nuclear factor-kappa B (NF-κB) in macrophages induced by lipopolysaccharide (LPS). Furthermore, treatment with isobavachin effectively suppressed the LPS-induced overproduction of NO, ROS, and neutrophils in a zebrafish inflammation model. This study demonstrated that bioavailable isobavachin-inhibited LPS-induced inflammatory responses via the MAPK and NF-κB signaling pathways, suggesting its potential as a modulatory agent in inflammatory diseases.

Four flavanone species, namely 5,7-dihydroxy-6-methyl-8-prenylflavanone, 5,7-dihydroxy-6-methyl-8-prenyl-4′-methoxy-flavanone, 5,7-dihydroxy-6-prenylflavanone, and 5-hydroxy-7-methoxy-6-prenylflavanone, with potent pharmacological properties were extracted from the leaves of *Eysenhardtia platycarpa*. These flavanones were then incorporated into nanoemulsion (NE) and polymeric nanoparticles (NPs) for topical application as novel anti-inflammatory formulations. The formulations exhibited anti-inflammatory properties, with the NE and NPs loaded with 5-hydroxy-7-methoxy-6-prenyl flavanone- showing the best results in release and skin permeation studies, and remarkable anti-inflammatory activity [81]. Figure 6 shows the structures of four prenylated flavanones from the Fabaceae family with anti-inflammatory activity.

The effects of several PF compounds (hedysarumines B, E, and G, and paratocarpins A-C, E, and F) on NO production in murine BV-2 microglial cells were investigated to determine their anti-inflammatory effects. The results revealed that hedysarumines B and G and paratocarpin F had significant inhibitory effects on NO production, with IC_50_ values of 5.12, 3.25, and 8.48 μM, respectively, which were comparable to those of dexamethasone, which was used as a positive control (9.47 μM). This suggests that these PF compounds could be as effective as dexamethasone, a commonly used anti-inflammatory drug. Hedysarumine E and paratocarpins B and E showed a weaker effect, with IC_50_ values between 10.33 and 18.18 μM, while paratocarpin A was ineffective (IC_50_ > 100 μM). The analysis revealed that the 2-(1-hydroxy-1-methylethyl)-2,3-di-hydrofuran ring present in hedysarumines B, E, and G and paratocarpin F played important role in the inhibition of NO production [82].

The ethanolic extracts from the roots, stems, and leaves of *Genista tridentata*, which are rich in flavonoids, have provided significant anti-inflammatory results. Three PFs, namely lupinifolin, mundulin, and 3-methoxymundulin, were isolated from roots and characterized. It was shown that root and stem extracts could remarkably reduce LPS-induced transcription of proinflammatory genes Il1b, Il6, and Ptgs2. This suggests that *G. tridentata* preparations could be a promising natural source for developing anti-inflammatory treatments. Accordingly, these findings emphasized the potential of *G. tridentata* preparations for anti-inflammatory treatments [83].

The purification of *Cullen corylifolium* revealed the presence of four prenylated chalcones: isobavachalcone, bavachalcone, bavachromene, and kanzonol B. These chalcones have a concentration-dependent inhibitory effect on LPS-activated microglia nitric oxide (NO) generation and PGE2. Prenylchalcones inhibited the protein and mRNA expression of cyclooxygenase-2 (COX-2) and iNOS in LPS-activated microglia. They also inhibited the degradation of inhibitor-κBα (I-κBα) and reduced the amount of nuclear factor κB (NF-κB) in LPS-stimulated BV-2 microglia. Thus, these prenylated chalcones could effectively treat neuroinflammatory diseases by modulating the expression of iNOS and COX-2 in activated microglial cells [84].

The anti-inflammatory effect of sophoraflavanone G and leachianone A was mainly mediated by their action on keratinocytes rather than macrophages. These PFs inhibited the production of proinflammatory mediators by targeting the MAPK, AP-1, and p65 signaling pathways. The combined prenylated and methoxylated flavanone, leachianone A, exhibited the highest skin transport ability. In addition, the cytokine/chemokine inhibitory effect of leachianone A was comparable to that of the commercial product betamethasone. Leachianone A also significantly reduced the severity of psoriasiform plaques without causing skin irritation or damage [85].

Compounds denominated SPF1 (2-[{3-hydroxy-2′,2-dimethyl-8-(3-methyl-2-butenyl)} chroman-6-yl]-7-hydroxy-8-(3-methyl-2-butenyl)-chroman-4-one) and SPF2 (2-[{2-(1-hydroxy-1-methylethyl)-7-(3-methyl-2-butenyl)-2′,3-dihydrobenzofuran}-5-yl]-7-hydroxy-8-(3-methyl-2-butenyl)chroman-4-one), two PFs found in the roots of *Sophora tonkinensis*, showed potent anti-inflammatory effects by binding to RXR. Both compounds reduced IL-1β mRNA and IL-6 mRNA levels in RAW264.7 cells stimulated with LPS and tumor necrosis factor-α. SPF1 also activated ATF3 mRNA and protein production, and reduced NF-κB translocation to the nuclei, significantly inhibiting proinflammatory cytokine production by RXR/LXR heterodimers. The effects of SPF1 could be partly due to the induction of ATF3, which can bind to the p65 subunit of NF-κB and reduce NF-κB transcription [86].

A recent research study found that the derivatives 8-prenyl-daidzein (8-PD) and 8-prenyl-genistein (8-PG) are more effective than daidzein and genistein in suppressing inflammatory responses in macrophages. The study also confirmed that 8-PD and 8-PG were able to suppress the activation of nuclear factor kappa B (NF-κB) and decrease the activation of ERK1/2, JNK, and p38 MAPK, resulting in suppressed phosphorylation of mitogen- and stress-activated kinase 1. Furthermore, the derivatives successfully suppressed the inflammatory responses induced by the medium containing hypertrophic adipocyte secretions and inhibited the proinflammatory secretion of C–C motif chemokine ligand 2 (CCL2) from adipose tissue of mice fed a long-term high-fat diet. These results suggested that 8-PD and 8-PG could effectively regulate macrophage activation under obesity conditions [87]. 

The information gathered suggests that certain prenylated flavonoids (PFs) and their derivatives have promising anti-inflammatory properties. Compounds such as sophoraflavanone G (SG), kushenol C (KC), isobavachin, and various flavanones isolated from different plants have shown anti-inflammatory effects via different mechanisms. These compounds have been shown to inhibit the production of inflammatory mediators such as ROS, interleukins, TNF-α, NF-kB, and NO by targeting key inflammatory signaling pathways. Additionally, they have exhibited regulatory effects on signaling molecules such as JNK/AP-1, PI3K/Akt, STAT, and NF-κB, contributing to their anti-inflammatory activity. In addition, incorporating certain flavanones into nanoemulsions and polymeric nanoparticles has shown promise in the development of novel anti-inflammatory formulations for topical application. Overall, the results emphasize the potential of PF compounds and their derivatives as natural candidates for treating inflammatory diseases. Further research into their mechanisms of action and their potential as modulatory agents in inflammatory diseases could pave the way for the development of new anti-inflammatory therapies. Table 4 summarizes the anti-inflammatory PF compounds from the Fabaceae family.

### 3.4. Anti-Diabetic Activity

Postprandial hyperglycemia is an important risk factor for the onset and development of type 2 diabetes. The inhibition of *α*-glucosidase can delay both the digestion and absorption of carbohydrates and thus suppress postprandial hyperglycemia. It should be emphasized that the prenylation of flavonoids increases their efficacy for *α*-glucosidase inhibition.

6-prenylquercetin from the leaves of *G. uralensis* inhibited *α*-glucosidase with an IC_50_ value of 3.7 μg mL^−1^ [88]. Hirtacoumaroflavonoside acted as a non-competitive *α*-glucosidase inhibitor, with an IC_50_ of 22 μM [88]. Hirtaflavonoside B showed *α*-glucosidase inhibitory activity, with an IC_50_ value of 71 μM but with a mixed non-competitive inhibitory pattern [89]. Morusinol and dioxycudraflavone A also exhibited *α*-glucosidase inhibitory activities, with IC_50_ values of 23.2 and 25.27 μM, respectively [32,90]. 4′,1″-dihydroxy-3′-methoxy-6,7-furanflavanone, and 5,3′,4′-trihydroxy-1″-methoxy-6,7-furanbavachalcone derived from the seeds of *P. corylifolia*, showed a remarkable inhibitory effect on *α*-glucosidase, with IC_50_ values of 53.1 μM and 90.3 μM, respectively, when compared to the positive control acarbose (IC_50_ of 214.5 μM) [91].

The inhibitory activity is more potent in prenylated isoflavonoids with a linear prenyl group than for the cyclized ones [92]. Moreover, adding the hydroxyl group to the prenyl group could significantly increase the inhibitory effect, while adding the OCH_3_ group decreases the inhibitory effect. Molecular docking analysis also supported the above assumption [92]. The structure of a prenylated isoflavonoid from the Fabaceae family with anti-diabetic activity is shown in Figure 7. 

The peroxisome proliferator-activated receptor-*γ* (PPAR-*γ*) belongs to the superfamily of nuclear hormone receptors that act as ligand-inducible transcription factors and modulate the expression of target genes involved in controlling glucose homeostasis, lipid metabolism, and inflammation [93].

Several PFs isolated from *P. corylifolia* were analyzed for their PPAR-*γ* agonist activity. Bavachin, bavachinin, 4′*-O*-methylbavachalcone, broussochalcone B, and corylifol A showed strong PPAR-*γ* agonist activation, ranging from 6.16-fold to 13.12-fold at a test concentration of 25 μM. Substitution of methoxyl at C-7 of the A-ring with hydroxyl, as with bavachinin and broussochalcone B, decreased PPAR-*γ* agonist activation compared to bavachinin and 4′*-O*-methylbavachalcone. This indicated that the methoxyl at the C-7 position of the A-ring was critical for PPAR-*γ* agonist activity. Isobavachin and isobavachalcone showed lower PPAR-*γ* agonist activity than bavachin and broussochalcone B, suggesting that the prenyl at the C-6 position of the A-ring may contribute to their agonist activities. In addition, 4′*-O*-methylbavachalcone and broussochalcone B exhibited lower PPAR-*γ* agonist activity than bavachin and bavachinin, showing that the C-ring structure is an essential determinant of PPAR-*γ* agonist activity and that the opening of the C-ring would lead to remarkably lower activity. The cyclization of the prenyl group on the A-ring and B-ring also significantly decreased PPAR-*γ* agonist activity [94]. 

The data can be summarized as follows: (1). Prenylation of flavonoids increases their potency in inhibiting α-glucosidase, with prenylated isoflavonoids with a linear prenyl group showing a stronger inhibitory activity. (2). The addition of a hydroxyl group to the prenyl group can significantly increase the inhibitory effect, while adding an OCH_3_ group reduces the inhibitory effect. (3). Several compounds isolated from *P. corylifolia* exhibit PPAR-γ agonist activity, with specific structural features such as methoxyl at the C-7 position of the A-ring being crucial for this activity. (4). The cyclization of the prenyl group on the A-ring and B-ring significantly reduces PPAR-γ agonist activity. This research provides valuable insight into the potential use of these compounds in the treatment of postprandial hyperglycemia and the modulation of PPAR-γ activity for controlling glucose homeostasis and lipid metabolism. The anti-diabetic activity of PFs found in Fabaceae plants is summarized in Table 5.

### 3.5. Anti-Alzheimer and Neuroprotective Activity

Neurodegenerative diseases (NDs) are a global public health problem whose incidence is increasing as people live longer [95]. Currently, palliative strategies are available for most NDs, but effective therapies for their cure are lacking. Flavonoids have been extensively studied for their multitarget behavior. Among their numerous biological activities, they have been reported to act at the central nervous system (CNS) level by exhibiting neuroprotective activity through various mechanisms of action. Current research shows that the nervous system is damaged by chronic oxidative stress and neuroinflammation, which alters communication between neurons and further impairs memory, cognition, and motor function. Stimulation of microglia is the main feature of neuroinflammation as it promotes the release of proinflammatory cytokines, which leads to neuronal death. All these processes contribute to the development of NDs, such as Alzheimer’s disease (AD), Parkinson’s disease, multiple sclerosis, and dementia [96].

Knowledge about the neuroprotective effects of natural products rich in phytochemicals, such as PFs, in preventing brain diseases, especially NDs, is rapidly expanding. However, more preclinical and clinical studies are still needed to fully understand the neuroprotective effects of PFs, including the prenylated compounds found in plants from the Fabaceae family.

In vivo evaluation of the neuroprotective activity of sophocarpine from *Sophora* genus revealed significant recovery rates from transient focal cerebral ischemia in sophocarpine-treated animals (5, 10, or 20 mg/kg) [97]. In *G. glabra*, a phenolic compound 2,2′,4′-trihydroxychalcone (TDC) has been isolated and has shown neuroprotective effects both in vitro and in vivo through decreasing BACE1 levels without affecting α- or γ-secretase levels [98]. Additionally, Barceló et al. (2017) carried out a preliminary toxicological assessment with (-)-(2S)-5,7, 2′,4′-tetrahydroxy-5′-(1‴,1‴-dimethylallyl)-8-prenylflavanone compounds from the roots of *Dalea* species in mice, which suggested an acceptable level of safety, encouraging future in vivo assays to validate neuroprotective activity, as well as to test its bioavailability, metabolic stability, safety, efficacy, and effective dose [65]. On the other hand, icariin, isolated from the Fabaceae plants, protects against Alzheimer’s disease by alleviating neuroinflammation oxidative stress and reducing potential risk factors for Alzheimer’s disease such as atherosclerosis in animal models [99].

The consumption of glyceollins (10 mg kg^−1^) from soybeans (*G. max*) prevented scopolamine-induced cognitive impairment in C57BL/6J wild-type mice. However, glyceollins did not affect memory improvement in mice [100]. Soybean glyceollins also regulated heme oxygenase-1 gene expression, which activated the Nrf2 signaling pathway and thus prevented glutamate-induced neuronal damage in primary cortical neurons [100]. As phytoestrogens, soybeans glyceollins have biological functions in the brain via estrogen receptors (ERs), as ERs are highly expressed in the brain [101]. The consumption of glyceollins increased the expression of genes related to neurogenesis, synaptic plasticity, and neuronal development in ovariectomized adult female mice. In addition, this consumption downregulated the expression of the gene for peptidylprolyl isomerase A, which affected inflammation and apoptosis in the CNS. Bamji et al. (2015) hypothesized that the beneficial effects of soybean glyceollins on the CNS could be mediated by ER-dependent and ER-independent mechanisms via specific genes [101].

Alzheimer’s disease (AD) is an age-related progressive disorder characterized by irreversible cognitive decline, memory loss, and behavioral disturbances [102]. It is an alarming fact that, according to the World Health Organization (WHO), three-quarters of the world population over the age of 60 will suffer from AD by 2025 [103]. To date, there is no consensus on the cause of AD. However, with increasing AD research, several hypotheses have been proposed based on the various pathophysiological factors observed, including amyloid beta peptide (Aβ) deposition, acetylcholine (ACh) deficits, tau protein aggregation, the dyshomeostasis of biometals, and oxidative stress [104]. PFs, an important class of small-molecule natural polyphenolic phytochemicals, can effectively treat AD.

Cholinesterases (ChEs) serve as key enzymes and are highly involved in the pathogenesis of AD. ChEs inhibitors may be more effective in the prevention and treatment of AD. The South American species *D. elegans* and *Dalea pazensis* are important sources of natural compounds of the prenylated flavanone type, such as: (−)-(2*S*)-2′,4′-dihydroxy-5′-(1‴,1‴-dimethylallyl)-8-prenylpinocembrin (8PP), (−)-(2*S*)-8-prenylpinocembrin (glabranin), (−)-(2*S*)-4′-hydroxy-2′-methoxy-5′-(1‴,1‴-dimethylallyl)-8-prenylpinocembrin (Me8PP), (−)-(2*S*)-3′,4′-dihydroxy-6,2′-diprenylpinocembrin (pazentin A), and (2*S*)-(−)-4′-hydroxy-2′-methoxy-5′-(1‴,1‴-dimethylallyl)-6-prenylpinocembrin (pazentin B) [95].

In an in vitro model of a culture of primary cerebellar granule neurons (CGN), these natural prenylated flavonones from the *Dalea* species were investigated for their neuroprotective potential. The compounds 8PP and glabranin showed neuroprotective effects against oxidative stress-induced death in both models: CGNs and (NGF)-differentiated PC12 cells. Structure–activity relationships were also reported. The results suggested that an 8-prenyl group on the A-ring in conjunction with an unsubstituted B-ring or a 2′,4′-dihydroxy-5′-dimethylallyl substitution led to the most effective flavanones. In addition, in silico studies were performed, and several putative targets in NDs were identified for compounds 8 PP and glabranin. Of these, the enzyme acetylcholinesterase was selected for validation in vitro. The results implied that the two above-mentioned prenylated flavanones could be helpful in the development and design of future strategies for the treatment of NDs [95].

Prenylated compounds from *Psoralea fructus*, bavachalcone, and bavachin showed potent inhibition of NO production similar to curcumin. These PFs demonstrated dose-dependent inhibitory effects, with IC_50_ values of 6.10 and 7.71 μM, respectively [105]. The obtained values were comparable to those of curcumin (IC_50_ of 6.61 μM). On the other hand, bavachinin and isobavachalcone showed moderate inhibition and achieved IC_50_ values of 19.32 and 27.06 μM, respectively [105]. The PF compounds with the strongest anti-Alzheimer’s and neuroprotective activity are shown in Figure 8.

Based on the literature, flavonoids, especially prenylated compounds from the Fabaceae family, which are rich in phytochemicals, have shown promising neuroprotective effects in the prevention and treatment of NDs. These compounds have demonstrated neuroprotective effects through different mechanisms of action, such as regulating heme oxygenase-1 gene expression, activating the Nrf2 signaling pathway, and influencing gene expression related to neurogenesis and synaptic plasticity. Specific compounds such as glyceollins from soybeans and prenylated flavanones from *Dalea* species have been reported to exhibit neuroprotective effects in in vitro models, which represents a potential for the development of future strategies for the treatment of NDs. Additionally, prenylated compounds from *P. fructus* have shown a potent inhibition of NO production, indicating their potential therapeutic value. These findings highlight the potential of flavonoids and prenylated compounds as promising candidates for the development of effective therapies for neurodegenerative diseases. However, further preclinical and clinical studies are needed to understand and fully exploit the neuroprotective effects of these compounds. Table 6 summarizes the anti-Alzheimer’s and neuroprotective activities of PFs from the Fabaceae family.

## 4. Conclusions

The findings described in this review suggest the emerging therapeutic potential of prenylated flavonoid compounds of the Fabaceae plants to protect against diseases, which is beneficial for human health. However, the scarcity of clinical trials with functional evidence makes it necessary to develop techniques and therapeutic doses to administer these compounds, since they can be modified when coming into contact with in vivo models. The PFs possess many bioactivities; PFs of the Fabaceae family can be developed as new drugs and supplements that will contribute to human health. For example, isobavachalcone and sophoraflavanone G could be good candidates for testing in animal models since they have shown the following biological activities: anti-inflammatory, anti-diabetic, anti-cancer, anti-Alzheimer, and neuroprotective activities. It is also crucial to emphasize the need for further research to evaluate the capacity of PFs from Fabaceae species, as very few compounds have been tested in animal models to verify in vitro results.

## Figures and Tables

**Figure 1 ijms-25-13036-f001:**
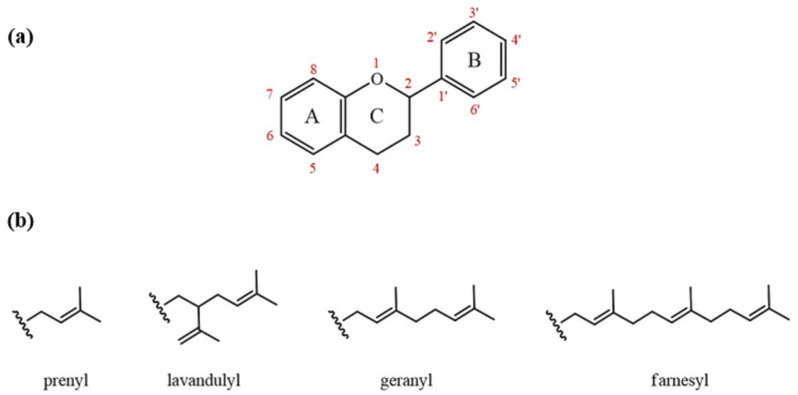
(**a**) Basic skeleton of flavonoids in plants. (**b**) Prenyl groups present in plants.

**Figure 2 ijms-25-13036-f002:**
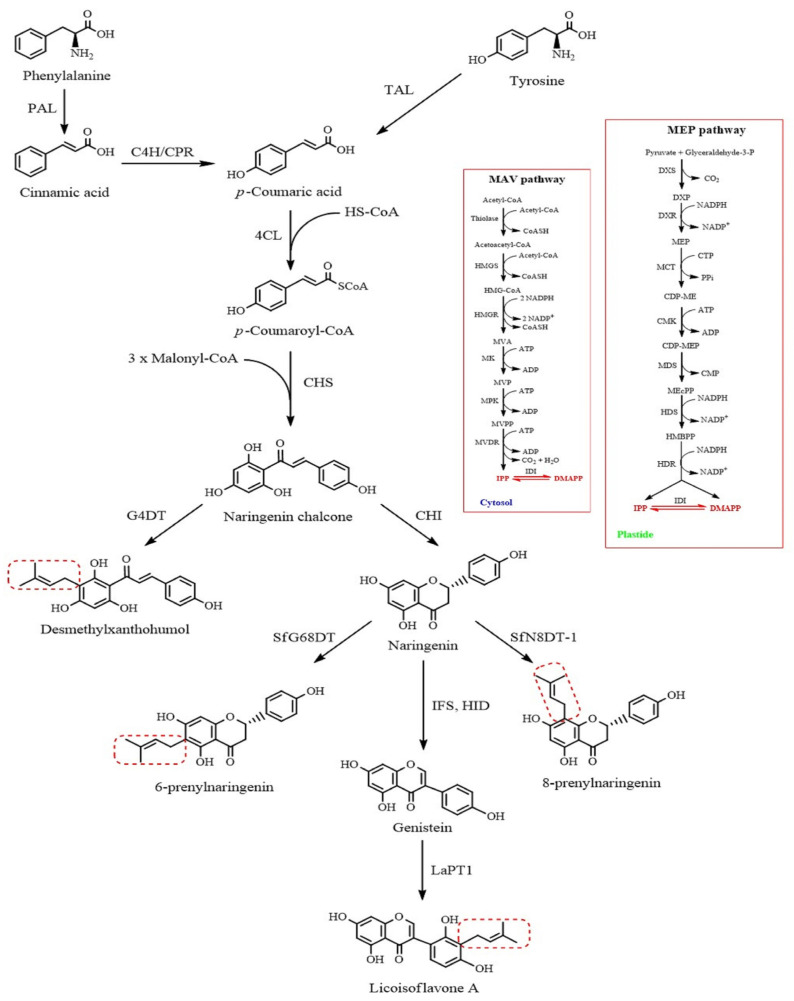
Metabolic pathway of prenylated flavonoids in plants. The dashed red boxes show the prenylation sites related to MVA and MEP pathways. MVA pathway: Acetyl-CoA, thiolase, acetoacetyl-CoA, HMG-CoA synthase (HMGS), 3-hydroxy-3-methyl-glutaryl-CoA (HMG-CoA), HMG-CoA reductase (HMGR), mevalonic acid (MVA), mevalonic acid 5-kinase (MK); mevalonic acid phosphate (MVP), mevalonic acid phosphate 5-kinase (MPK); mevalonic acid 5-diphosphate (MVPP), mevalonic acid diphosphate decarboxylase (MVD); isopentenyl diphosphate (IPP), isopentenyl diphosphate isomerase (IDI), dimethylallyl diphosphate (DMAPP). MEP pathway: Piruvate + Glyceraldehyde-3-phosphate, 1-Deoxy-D-xylulose-5-phosphate synthase (DXS), 1-Deoxy-D-xylulose-5-phosphate (DXP), 1-Deoxy-D-xylulose-5-phosphate reductoisomerase (DXR), 2C-Methyl-D-erythritol-4-phosphate (MEP), MEP cytidyl transferase (MCT), 4-(Cytidine 5′-diphospho)-2C-methyl-D-erythritol (CDP-ME), Cytidyl MEP kinase (CMK), 4-(Cytidine 5′-diphospho)-2C-methyl-D-erythritol-2-phosphate (CDP-MEP), MEP-2,4-cyclodiphosphate synthase (MDS), 2C-methyl-D-erythritol-2,4-cyclodiphosphate (MEcPP), (*E*)-4-Hydroxy-3-methylbut-2-enyl diphosphate synthase (HDS), (*E*)-4-Hydroxy-3-methylbut-2-enyl diphosphate (HMBPP), (*E*)-4-Hydroxy-3-methylbut-2-enyl diphosphate reductase (HDR), isopentenyl diphosphate (IPP), Dimethylallyl diphosphate (DMAPP), Isopentenyl diphosphate isomerase (IDI). Next, endogenous FPP synthase converts DMAPP and IPP into more extended prenyl donors, such as geranyl diphosphate and farnesyl diphosphate. Additionally, flavonoid prenyltransferase enzymes (SfN8DT-1, SfG68DT, LaPT1) can produce diverse prenylated flavonoids by condensation with prenyl donors. This is the most crucial stage in PF compound production in plants.

**Figure 3 ijms-25-13036-f003:**
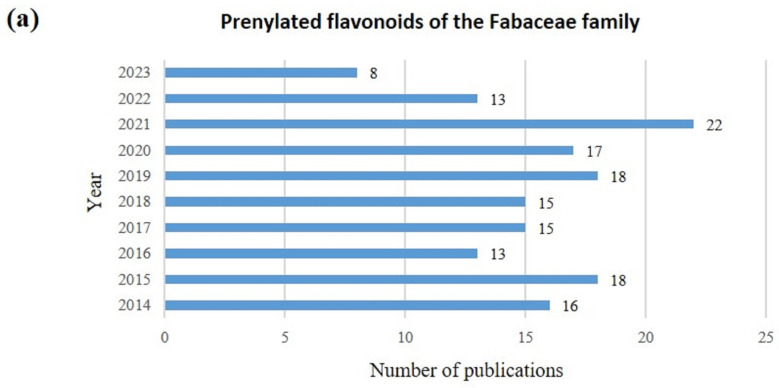
(**a**) The number of PubMed and Web of Knowledge articles on PF compounds of the Fabaceae family (2014–2023). (**b**) PF compounds by Fabaceae species and biological activities (each color of the circle shows the number of PF compounds identified per Fabaceae species and biological activity). ND‡ corresponds to unidentified Fabaceae plant species.

**Figure 4 ijms-25-13036-f004:**
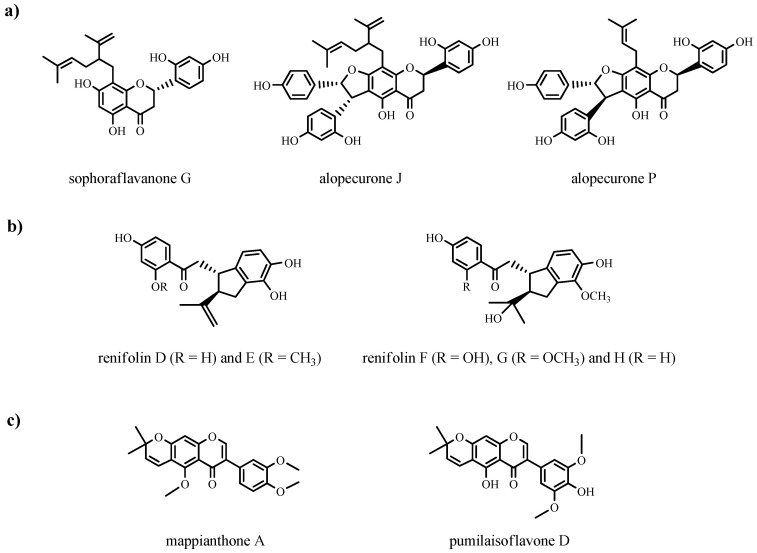
Some structures of PF compounds of the Fabaceae family with anti-cancer activity: (**a**) Prenylated flavostilbenes (**b**) Prenylated chalcones, (**c**) Prenylated isoflavones.

**Figure 5 ijms-25-13036-f005:**
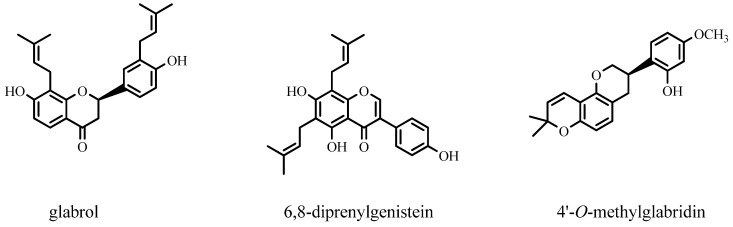
Structures of prenylated flavonoids from the Fabaceae family with high activity against MRSA.

**Figure 6 ijms-25-13036-f006:**
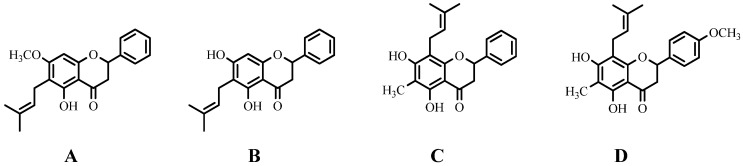
Structures of four prenylated flavanones from the Fabaceae family with anti-inflammatory activity formulated as nanoemulsion or polymeric nanoparticles for topical use. 5-hydroxy-7-methoxy-6-prenylflavanone (**A**); 5,7-dihydroxy-6-prenylflavanone (**B**); 5,7-dihydroxy-6-methyl-8-prenylflavanone (**C**); and 5,7-dihydroxy-6-methyl-8-prenylflavanone (**C**); and 5,7-dihydroxy-6-methyl-8-prenyl-4′-methosyflavanone (**D**).

**Figure 7 ijms-25-13036-f007:**
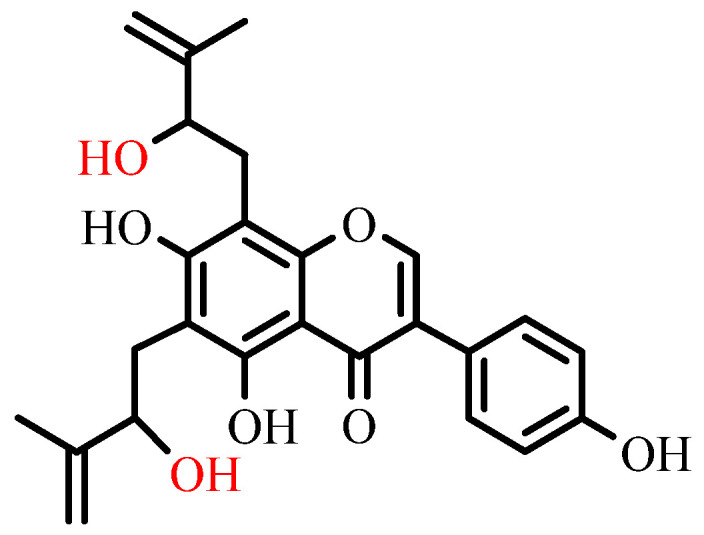
Structure of a prenylated isoflavonoid from the Fabaceae family with anti-diabetic activity. Hydroxyl groups that enhance the inhibitory effect are shown in red.

**Figure 8 ijms-25-13036-f008:**
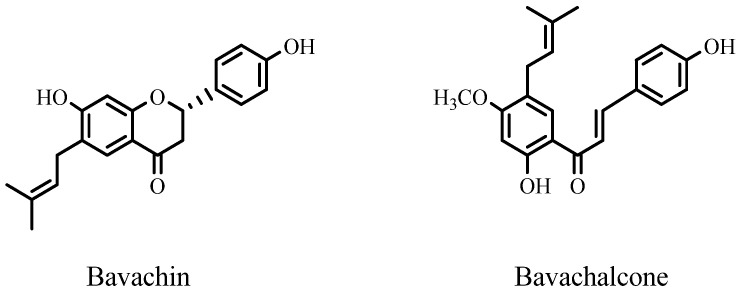
Structures of prenylated flavonoids from the Fabaceae family with anti-Alzheimer’s and neuroprotective activity.

**Table 1 ijms-25-13036-t001:** Biological multiactivities of some PF compounds of the Fabaceae family.

PF Compounds	ACA	AIA	AMA	ADA	AAA
Abyssinone V-4′-methyl ether	✓	✓			
Bavachalcone		✓			✓
Bavachin				✓	✓
Bavachinin				✓	✓
Hedysarumine	✓	✓			
Isobavachalcone		✓		✓	✓
Isobavachin		✓		✓	
Kanzonol C	✓	✓			
Lupinifolin		✓	✓		
Mundulin		✓	✓		
Sophoraflavanone G	✓	✓	✓		

ACA (anti-cancer activity); AIA (anti-inflammatory activity); AMA (antimicrobial activity); ADA (anti-diabetic activity); and AAA (anti-Alzheimer and neuroprotective activity). The biological activity of each compound is shown with a check (✓).

**Table 2 ijms-25-13036-t002:** Anti-cancer activity of PF compounds of the Fabaceae family.

Compound	Cell Line (IC_50_ *)	Species	Reference
2′,3-hydroxy-4,4′-dimethoxychalcone	A549: IC_50_ = 3.5 μM PC3: IC_50_ = 6.8 μM MCF-7: IC_50_ = 6.25 μM	*Desmodium podocarpum*	[40]
2′,4-hydroxy-3,4′-dimethoxychalcone	SHSY5Y: IC_50_ = 3.8 μM PC3: IC_50_ = 3.6 μM NB4:IC_50_ = 4.2 μM	*Desmodium podocarpum*	[40]
3″,4″-dihydrothonningine C	HL-60: IC_50_ = 0.18 μM SMMC-7721: IC_50_ = 0.18 μM A-549: IC_50_ = 0.18 μM MCF-7: IC_50_ = 0.18 μM SW480: IC_50_ = 18.76 μM	*Glycyrrhiza uralensis*	[35]
3′-formylalpinumi soflavone	HL-60: IC_50_ = 1.27 ± 0.10 μM SMMC-7721: IC_50_ = 2.86 ± 0.11 μM A-549: IC_50_ = 6.38 ± 0.09 μM MCF-7: IC_50_ = 12.68 ± 0.07 μM SW480: IC_50_ = 2.66 ± 0.08	*Mappianthus iodoides*	[33]
3-hydroxy-4-methoxylonchocarpin	K562: IC_50_ = 2.4 µg/mL	*Millettia pachycarpa*	[39]
4′-hydroxy-5,7-dimethoxy-6-(3-methyl-2-butenyl)-isoflavone	HL-60: IC_50_ = 0.42 μM SMMC-7721: IC_50_ = 0.45 μM A-549: IC_50_ = 0.51 μM MCF-7: IC_50_ = 1.62 μM SW480: IC_50_ = 8.48 μM	*Glycyrrhiza uralensis*	[36]
5,7,3′-trihydroxy-2′-(3-methylbut-2-enyl)-4′,5′-(3,3-dimethylpyrano) isoflavone	MCF-7: IC_50_ = 4.8 μM A549: IC_50_ = 6.8 μM HepG2: IC_50_ = 24.8 μM	*Flemingia philippinensis*	[36]
5,7-dihydroxy-5′-prenyl-[2″,2″-(3″-hydroxy)-dimethylpyrano]-(5″,6″:3′,4′)favanone	H4IIE: EC_50_ = 14.7 µM	*Erythrina addisoniae*	[37]
5-hydroxysophoranone	HeLa: IC_50_ = 12 µM SK-MEL-5: IC_50_ = 7.7 µM	*Maackia amurensis*	[32]
5-*O*-methyl-2′-methoxy-3′-methyl alpinumisoflavone	HL-60: IC_50_ = 0.98 ± 0.06 μM SMMC-7721: IC_50_ = 3.25 ± 0.09 μM A-549: IC_50_ = 6.29 ± 0.13 μM MCF-7: IC_50_ = 7.56 ± 0.12 μM SW480: IC_50_ = 3.27 ± 0.10 μM	*Mappianthus iodoides*	[33]
6,8-diprenyleriodictyol	MRC-5: IC_50_ = 8.0 µM KB: IC_50_ = 8.5 µM	*Derris ferruginea*	[36]
Abyssinone V	HeLa: IC_50_ = 16–36 µM	*Maackia amurensis*	[32]
Addisoniafavanones I	H4IIE: EC_50_ = 5.25 µM	*Erythrina addisoniae*	[37]
Addisoniafavanones II	H4IIE: EC_50_ = 8.5 µM	*Erythrina addisoniae*	[37]
Alopecurone J	MCF-7: IC_50_ = 8.25 µM MDA-MB-231: IC_50_ = 16.36 µM NIH/3T3: IC_50_ = 14.67 µM	*Sophora pachycarpa*	[27]
Alopecurone J	HeLa: IC_50_ = 9.97 μM HCT116: IC_50_ = 10.17 μM A2780: IC_50_ = 12.45 μM A549: IC_50_ = 30.91 μM	*Glycyrrhiza uralensis*	[36]
Alopecurone P	ND	*Sophora pachycarpa*	[27]
Artonin ZA-1	NB4, A549, SHSY5Y, PC3: IC_50_ = 5.8–10 μM.	*Desmodium podocarpum*	[40]
Artonin ZA-2	NB4, A549, SHSY5Y, PC3: IC_50_ = 5.8–10 μM.	*Desmodium podocarpum*	[40]
Asthonningine B	HL-60: IC_50_ = 0.18 μM SMMC-7721: IC_50_ = 0.18 μM A-549: IC_50_ = 0.18 μM MCF-7: IC_50_ = 0.18 μM SW480: IC_50_ = 18.76 μM	*Glycyrrhiza uralensis*	[35]
Auriculasin	PC-3: GI_50_ = 8.33 μM	*Flemingia philippinensis*	[50]
*cis*-(6a*β*,12a*β*)-hydroxyrotenone	HT-29: IC_50_ = 0.1 μM CCD-112CoN: IC_50_ > 100 μM	*Indigofera spicata*	[45]
Dihydrorobustone methyl ether	HL-60: IC_50_ = 4.83 ± 0.12 µM SMMC-7721: IC_50_ = 1.56 ± 0.07 µM A-549: IC_50_ = 3.29 ± 0.08 µM MCF-7: IC_50_ = 2.67 ± 0.11 µM SW480: IC_50_ = 5.36 ± 0.13 µM	*Mappianthus iodoides*	[33]
Dorsmanine I	KB: IC_50_ = 23.8 µM	*Derris ferruginea*	[36]
Eriosematin	PC-3: GI_50_ > 100 μM	*Flemingia philippinensis*	[36]
Erythraddison A	HL-60: IC_50_ = 2.69 ± 0.08 µM SMMC-7721: IC_50_ = 3.98 ± 0.13 µM A-549: IC_50_ = 1.69 ± 0.07 µM MCF-7: IC_50_ = 5.72 ± 0.16 µM SW480: IC_50_ = 8.63 ± 0.12 µM	*Mappianthus iodoides*	[33]
Ficucaricone D	HL-60: IC_50_ = 0.42 μM SMMC-7721: IC_50_ = 0.45 μM A-549: IC_50_ = 0.51 μM MCF-7: IC_50_ = 1.62 μM SW480: IC_50_ = 8.48 μM	*Glycyrrhiza uralensis*	[36]
Ficucaricones A	HL-60: IC_50_ = 0.18 μM SMMC-7721: IC_50_ = 0.18 μM A-549: IC_50_ = 0.18 μM MCF-7: IC_50_ = 0.18 μM SW480: IC_50_ = 18.76 μM	*Glycyrrhiza uralensis*	[35]
Ficucaricones B	HL-60: IC_50_ = 0.18 μM SMMC-7721: IC_50_ = 0.18 μM A-549: IC_50_ = 0.18 μM MCF-7: IC_50_ = 0.18 μM SW480: IC_50_ = 18.76 μM	*Glycyrrhiza uralensis*	[35]
Flemiphilippinin G	MCF-7: IC_50_ = 4.8 μM A549: IC_50_ = 5.8 μM HepG2: IC_50_ = 24.8 μM	*Flemingia philippinensis*	[36]
Flemiphilippinone C	PC-3: GI_50_ = 14.12 μM Bel-7402: GI_50_ = 1.91 μM CaEs-17: GI_50_ = 1.97 μM GI50: GI_50_ = 2.58 μM	*Flemingia philippinensis*	[43]
Gancaonin G	SW480: IC_50_ = 9.84 μM	*Glycyrrhiza uralensis*	[36]
Gancaonin N	HL-60: IC_50_ = 0.42 μM SMMC-7721: IC_50_ = 0.45 μM A-549: IC_50_ = 0.51 μM MCF-7: IC_50_ = 1.62 μM SW480: IC_50_ = 8.48 μM	*Glycyrrhiza uralensis*	[35]
Glyurallin A	W480: IC_50_ = 10.86 μM	*Glycyrrhiza uralensis*	[35]
Hedysarumine F	A549: IC_50_ = 7.79 μM	*Hedysarum gmelinii*	[42]
Indicanine A	HL-60: IC_50_ = 0.18 μM SMMC-7721: IC_50_ = 0.18 μM A-549: IC_50_ = 0.18 μM MCF-7: IC_50_ = 0.18 μM SW480: IC_50_ = 18.76 μM	*Glycyrrhiza uralensis*	[35]
Indicanine B	HL-60: IC_50_ = 0.18 μM SMMC-7721: IC_50_ = 0.18 μM A-549: IC_50_ = 0.18 μM MCF-7: IC_50_ = 0.18 μM SW480: IC_50_ = 18.76 μM	*Glycyrrhiza uralensis*	[35]
Indicanine E	HL-60: IC_50_ = 2.45 ± 0.09 µM SMMC-7721: IC_50_ = 4.28 ± 0.16 µM A-549: IC_50_ = 6.35 ± 0.12 µM MCF-7: IC_50_ = 4.27 ± 0.10 µM SW480: IC_50_ = 1.32 ± 0.06 µM	*Mappianthus iodoides*	[33]
Isomaackiafavanone A	HeLa: IC_50_ = 16–36 µM	*Maackia amurensis*	[32]
Isomaackiafavanone B	HeLa: IC_50_ = 16–36 µM	*Maackia amurensis*	[32]
Kanzonol C	A549: IC_50_ = 9.67 μM HCT116: IC_50_ = 8.85 μM	*Hedysarum gmelinii*	[42]
Licofavonol	SW480: IC_50_ = 14.53 µM HeLa: IC_50_ = 8.23 µM	*Glycyrrhiza uralensis*	[36]
Lonchocarpol A	MRC-5: IC_50_ = 6.2 µM	*Derris ferruginea*	[36]
Maackiafavanone	HeLa: IC_50_ = 8.2 µM SK-MEL-5: IC_50_ = 6.5 µM	*Maackia amurensis*	[29]
Maakiaflavanone A	A375S2: IC_50_ = 35.1 ± 0.8 µM Hela: IC_50_ = 27.7 ± 1.1 µM MCF-7: IC_50_ = 39.2 ± 0.9 µM Hep G2: IC_50_ = 40.3 ± 0.7 µM SK-MEL-5: IC_50_ = 33 µM	*Maackia amurensis*	[29]
Maakiaflavanone B	A375S2: IC_50_ = 7.8 ± 1.4 µM Hela: IC_50_ = 36.8 ± 1.0 µM MCF-7: IC_50_ = 16.8 ± 1.2 µM Hep G2: IC_50_ = 37.4 ± 0.9 µM SK-MEL-5: IC_50_ = 8.8 µM	*Maackia amurensis*	[29]
Mappianthone A	HL-60: IC_50_ = 0.16 ± 0.05 µM SMMC-7721: IC_50_ = 1.23 ± 0.09 µM A-549: IC_50_ = 0.68 ± 0.06 µM MCF-7: IC_50_ = 0.39 ± 0.04 µM SW480: IC_50_ = 1.63 ± 0.12 µM	*Mappianthus iodoides*	[33]
Nitidulin	KB: IC_50_ = 3.47 ± 0.09 µM Hela-S3: IC_50_ = 5.17 ± 0.11 µM HepG2: IC_50_ = 7.19 ± 0.36 µM HT-29: IC_50_ = 9.76 ± 0.45 µM	*Dalbergia velutina*	[28]
Pumilaisoflavone D	HL-60: IC_50_ = 3.02 ± 0.11 µM SMMC-7721: IC_50_ = 7.29 ± 0.13 µM A-549: IC_50_ = 4.20 ± 0.15 µM MCF-7: IC_50_ = 8.53 ± 0.17 µM SW480: IC_50_ = 3.28 ± 0.12 µM	*Mappianthus iodoides*	[33]
Renifchalcone A	A549: IC_50_ = 5.8 µM MCF7: IC_50_ = 6.2 µM	*Desmodium renifolium*	[51]
Renifolin D	NB4: IC_50_ = 6.3 µM A549: IC_50_ > 10 µM SHSY5Y: IC_50_ > 10 µM PC3: IC_50_ > 5.8 µM MCF7: IC_50_ > 10 µM	*Desmodium renifolium*	[32]
Renifolin E	NB4: IC_50_ = 6.4 µM A549: IC_50_ = 2.8 µM SHSY5Y: IC_50_ = 8.6 µM PC3: IC_50_ > 10 µM MCF7: IC_50_ = 5.5 µM	*Desmodium renifolium*	[32]
Renifolin F	NB4: IC_50_ = 3.8 µM A549: IC_50_ = 2.2 µM SHSY5Y: IC_50_ = 4.5 µM PC3: IC_50_ = 7.6 µM MCF7: IC_50_ = 8.7 µM	*Desmodium renifolium*	[32]
Renifolin G	NB4: IC_50_ = 6.5 µM A549: IC_50_ = 4.8 µM SHSY5Y: IC_50_ = 7.2 µM PC3: IC_50_ = 9.7 µM MCF7: IC_50_ = 8.8 µM	*Desmodium renifolium*	[32]
Renifolin H	NB4: IC_50_ > 10 µM A549: IC_50_ = 5.9 µM SHSY5Y: IC_50_ = 8.6 µM PC3: IC_50_ > 10 µM MCF7: IC_50_ = 9.7 µM	*Desmodium renifolium*	[32]
Renifolins C	NB4: IC_50_ = 6.4 μM. PC3: IC_50_ = 8.5 μM.	*Desmodium renifolium*	[32,41]
Renifolins D	A549: IC_50_ = 2.2 μM PC3: IC_50_ = 2.7 μM MCF-7: IC_50_ = 9.7 μM	*Desmodium renifolium*	[32,41]
Renifolins E	A549: IC_50_ = 2.2 μM	*Desmodium renifolium*	[32,41]
Renifolins F	PC3: IC_50_ = 2.7 μM	*Desmodium renifolium*	[32,41]
Renifolins G	MCF-7: IC_50_ = 9.7 μM	*Desmodium renifolium*	[32,41]
Renifolins H	A549: IC_50_ = 2.2 μM	*Desmodium renifolium*	[32,41]
Robustone methyl ether	HL-60: IC_50_ = 0.89 ± 0.06 µM SMMC-7721: IC_50_ = 1.97 ± 0.11 µM A-549: IC_50_ = 2.01 ± 0.07 µM MCF-7: IC_50_ = 3.04 ± 0.08 µM SW480: IC_50_ = 0.53 ± 0.05 µM	*Mappianthus iodoides*	[33]
Rotenone	HT-29: IC_50_ = 0.1 μM CCD-112CoN: IC_50_ > 100 μM	*Indigofera spicata*	[45]
Sanjoseolide	PC-3: IC_50_ = 5.2 µM DU 145: IC_50_ = 8.3 µM	*Dalea frutescens*	[38]
Sanjuanolide	PC-3: IC_50_ = 3.2 µM DU 145: IC_50_ = 3.6 µM	*Dalea* *frutescens*	[38]
Sophoflavescenol	A549: IC_50_ = 69.9 µM LLC: IC_50_ = 38.1 µM HL-60: IC_50_ = 12.5 µM MCF-7: IC_50_ > 150 µM	*Sophora flavescens*	[17]
Sophoraflavanone G	MCF-7: IC_50_ = 40.86 µM MDA-MB-231: IC_50_ = 52.63 µM NIH/3T3: IC_50_ = 86.5 µM	*Sophora pachycarpa*	[27]
Tephrosin	HT-29: IC_50_ = 0.1 μM 697: IC_50_ = 9.0 μM	*Indigofera spicata*	[45]
Thonningine A	HL-60: IC_50_ = 0.18 μM SMMC-7721: IC_50_ = 0.18 μM A-549: IC_50_ = 0.18 μM MCF-7: IC_50_ = 0.18 μM SW480: IC_50_ = 18.76 μM	*Glycyrrhiza uralensis*	[35]
Viridiflorin	HL-60: IC_50_ = 0.42 μM SMMC-7721: IC_50_ = 0.45 μM A-549: IC_50_ = 0.51 μM MCF-7: IC_50_ = 1.62 μM SW480: IC_50_ = 8.48 μM	*Glycyrrhiza uralensis*	[36]

* (unless otherwise specified); IC_50_ (half-maximal inhibitory concentration); ND (not determined); EC_50_ (median effective concentration); GI_50_ (concentration to inhibit the growth of cancer cells by 50%).

**Table 3 ijms-25-13036-t003:** Antimicrobial activity of PF compounds of the Fabaceae family.

Compound	Target(s) (MIC *)	Species	Reference
(2*S*)-5′-(2-methylbut-3-en-2-yl)-8-(3-methylbut-2-en-1-yl)-5,7,2′,4′-tetrahydroxyflavanone	*Streptococcus mutans*, *Bacillus cereus*, OSSA, and ORSA, 2.3–3.4 μg mL^−1^	*Dalea searlsiae*	[59]
(3*S*)-licoricidin	*Staphylococcus aureus*, *Proteus vulgaris*, *Salmonella typhimurium*, *Escherichia coli*, *Pseudomonas aeruginosa,* and *Candida albicans,* 4–62 μg mL^−1^	*Glycyrrhiza iconica*	[55]
(*E*)-5-hydroxytephrostachin	*Plasmodium falciparum* D6 strain 1.7 ± 0.1 μM	*Tephrosia purpurea* subsp. *leptostachya*	[61]
10-(γ,γ-dimethylallyl)-3,9,13-trihydroxy-6,12-metano-6H,12H-dibenzo[b,f][1,5]dioxocin	*Zygosaccharomyces rouxii* F51 >125.0 μg mL^−1^	*Desmodium caudatum*	[63]
2-(3-methyl-2-butenyl)-3,5,4′-trihydroxy-bibenzyl	*Staphylococcus aureus* and *Staphylococcus epidermidis,* 12.5–50.0 μg mL^−1^	*Glycyrrhiza inflata*	[64]
2′,4′-dihydroxy-5′-(1‴, 1‴-dimethylallyl)-8-prenylpinocembrin	*Candida albicans* 150 μM	*Dalea elegans*	[65]
2′,4′,4,6′-tetrahydroxy-3,3′-diprenyldihydrochalcone	*Bacillus subtilis* 5.5 μM	*Eriosema montanum*	[56]
2′,4′,5,6′-tetrahydroxy-4-methoxy-3,3′-diprenyldihydrochalcone	*Bacillus subtilis* 8.9 μM	*Eriosema montanum*	[56]
2″,2″-dimethylpyran-(5″,6″:6,7)-5,2′,4′-trihydroxy-(2*R*,3*R*)-dihydroflavonone	*Zygosaccharomyces rouxii* F51 31.3 μg mL^−1^	*Desmodium caudatum*	[63]
2″,2″-dimethylpyran-(5″,6″:6,7)-5,4′-dihydroxy-4′-methoxy-flavonol	*Zygosaccharomyces rouxii* F51 15.6 μg mL^−1^	*Desmodium caudatum*	[63]
2″,2″-dimethylpyran-(5″,6″:7,8)-5,2′,4′-trihydroxy-(2R,3*R*)-dihydroflavonone	*Zygosaccharomyces rouxii* F51 62.5 μg mL^−1^	*Desmodium caudatum*	[63]
2″,2″-dimethylpyran-(5″,6″:7,8)-5,2′,4′-trihydroxy-6-methyl-flavone	*Zygosaccharomyces rouxii* F51 125.0 μg mL^−1^	*Desmodium caudatum*	[63]
2″,2″-dimethylpyran-(5″,6″:7,8)-5,2′,4′-trihydroxyflavonol	*Zygosaccharomyces rouxii* F51 31.3 μg mL^−1^	*Desmodium caudatum*	[63]
2″,2″-dimethylpyran-(5″,6″:7,8)-5,2′-dihydroxy-4′-methoxy-(2*R*,3*R*)-dihydroflavonol	*Zygosaccharomyces rouxii* F51 7.8 μg mL^−1^	*Desmodium caudatum*	[63]
2″,2″-dimethylpyran-(5″,6″:7,8)-5,3′,4′-trihydroxy-6-methyl-(2*R*)-flavanone	*Zygosaccharomyces rouxii* F51 31.3 μg mL^−1^	*Desmodium caudatum*	[63]
2″,2″-dimethylpyran-(5″,6″:7,8)-5,3′,4′-trihydroxy-6-methyl-flavone	*Zygosaccharomyces rouxii* F51 >125.0 μg mL^−1^	*Desmodium caudatum*	[63]
2″,2″-dimethylpyran-(5″,6″:7,8)-5,3′-dihydroxy-4′-methoxy-(2*R*,3*R*)-dihydroflavonol	*Zygosaccharomyces rouxii* F51 31.3 μg mL^−1^	*Desmodium caudatum*	[63]
2″,2″-dimethylpyran-(5″,6″:7,8)-5,4′-dihydroxy-(2*R*,3*R*)-dihydroflavonol	*Zygosaccharomyces rouxii* F51 15.6 μg mL^−1^	*Desmodium caudatum*	[63]
2″,2″-dimethylpyran-(5″,6″:7,8)-5,4′-dihydroxyflavonol	*Zygosaccharomyces rouxii* F51 125.0 μg mL^−1^	*Desmodium caudatum*	[63]
2″-hydroxymethyl-2″-methylpyran-(5″,6″:7,8)-5,4′-(2*R*,3*R*)-dihydroxydihydroflavonol	*Zygosaccharomyces rouxii* F51 125.0 μg mL^−1^	*Desmodium caudatum*	[63]
3′-hydroxy-4′-*O*-methyl-glabridin	*Staphylococcus aureus* and MRSA, 44 μM	ND ^‡^	[57]
3′-methylorobol	*Staphylococcus aureus*, *Staphylococcus epidermidis* and *Bacillus subtilis,* 32–64 μg mL^−1^	*Millettia extensa*	[66]
4′-*O*-methyl-glabridin	*Staphylococcus aureus* and MRSA, 30 μM	ND ^‡^	[57]
5,7,3′,4′ -tetrahydroxy-6-(3′,3′-dimethylallyl)-flavanone	*Staphylococcus aureus* and *Staphylococcus epidermidis,* 12.5–50.0 μg mL^−1^	*Glycyrrhiza inflata*	[64]
5,7,3′,4′-tetrahydroxy-6,8-diprenylisoflavone	*Micrococcus luteus*, *Streptococcus mutans*, *Bacillus cereus*, and *Staphylococcus aureus,* 16–32 μg mL^−1^	*Millettia extensa*	[67]
5,7,3′,4′-tetrahydroxy-8-(3′,3′-dimethylallyl)-flavanone	*Staphylococcus aureus* and *Staphylococcus epidermidis,* 12.5–25.0 μg mL^−1^	*Glycyrrhiza inflata*	[64]
5,7,3′-trihydroxy-4′-methoxy-8-prenylisoflavone	MRSA, *Enterococcus faecium*, *Candida albicans*, *Cryptococcus neoformans,* and *Trichophyton rubrum,* IC_50_ 6.8–26.9 μM	*Vatairea guianensis*	[54]
5,7,4′-trihydroxy-(2*R*)-flavanone	*Zygosaccharomyces rouxii* F51 125.0 μg mL^−1^	*Desmodium caudatum*	[63]
5,7,4′-trihydroxy-(2*R*,3*R*)-dihydroflavonol	*Zygosaccharomyces rouxii* F51 125.0 μg mL^−1^	*Desmodium caudatum*	[63]
5,7,4′-trihydroxyflavonol	*Zygosaccharomyces rouxii* F51 15.6 μg mL^−1^	*Desmodium caudatum*	[63]
5,7-dihydroxy-6-(3′,3′-dimethylallyl)-flavanone	*Staphylococcus aureus* and *Staphylococcus epidermidis,* >100 μg mL^−1^	*Glycyrrhiza inflata*	[64]
5,7-dihydroxy-8-(3′,3′-dimethylallyl)-flavanone	*Staphylococcus aureus* and *Staphylococcus epidermidis* 25 μg mL^−1^	*Glycyrrhiza inflata*	[64]
5-deoxy-3′-prenylbiochanin A	MRSA, *Salmonella enterica subsp. enterica*, *Escherichia coli,* and *Candida albicans,* >95.1 μM	*Erythrina sacleuxii*	[60]
5-hydroxy-7-methoxy-4′-*O*-(3-methylbut-2-enyl)isoflavone	*Staphylococcus epidermidis* and *Staphylococcus aureus,* 128 μg mL^−1^	*Millettia extensa*	[67]
6-(γ,γ-dimethylallyl)-5,7,2′,4′-tetrahydroxy-(2*R*,3*R*)-dihydroflavonol	*Zygosaccharomyces rouxii* F51 62.5 μg mL^−1^	*Desmodium caudatum*	[63]
6-(γ,γ-dimethylallyl)-5,7,2′,4′-tetrahydroxyflavonol	*Zygosaccharomyces rouxii* F51 62.5 μg mL^−1^	*Desmodium caudatum*	[63]
6-(γ,γ-dimethylallyl)-5,7,4′-trihydroxyflavonol	*Zygosaccharomyces rouxii* F51 62.5 μg mL^−1^	*Desmodium caudatum*	[63]
6,8-diprenylgenistein	*Staphylococcus aureus* and MRSA, 23 μM	ND ^‡^	[57]
6-prenyl daidzein	*Listeria monocytogenes* and *Escherichia coli,* 21–28 μM	*Glycine max* (L.) Merrill	[68]
6-prenylnaringenin	*Staphylococcus aureus* and MRSA, 110 μM	ND ^‡^	[57]
6-prenylnaringenin	*Staphylococcus aureus* and *Staphylococcus epidermidis,* >100 μg mL^−1^	*Glycyrrhiza inflata*	[64]
7,4′-dihydroxy-(2*R*,3*R*)-dihydroflavonol	*Zygosaccharomyces rouxii* F51 125.0 μg mL^−1^	*Desmodium caudatum*	[63]
7,4′-dihydroxy-8,3′-diprenylflavone	MRSA 20.5 μM	*Erythrina sacleuxii*	[60]
7,4′-di-*O*-prenylgenistein	*Staphylococcus epidermidis* and *Staphylococcus aureus,* 128 μg mL^−1^	*Millettia extensa*	[67]
8-(γ,γ-dimethylallyl)-5,7,2′,4′-tetrahydroxy-(2*R*)-flavanone	*Zygosaccharomyces rouxii* F51 62.5 μg mL^−1^	*Desmodium caudatum*	[63]
8-(γ,γ-dimethylallyl)-5,7,2′,4′-tetrahydroxy-(2*R*,3*R*)-dihydroflavonol	*Zygosaccharomyces rouxii* F51 31.3 μg mL^−1^	*Desmodium caudatum*	[63]
8-(γ,γ-dimethylallyl)-5,7,2′,4′-tetrahydroxy-(2*R*,3*S*)-dihydroflavonol	*Zygosaccharomyces rouxii* F51 62.5 μg mL^−1^	*Desmodium caudatum*	[63]
8-(γ,γ-dimethylallyl)-5,7,2′-trihydroxy-4′-methoxy-(2*R*)-flavanone	*Zygosaccharomyces rouxii* F51 125.0 μg mL^−1^	*Desmodium caudatum*	[63]
8-(γ,γ-dimethylallyl)-5,7,4′-trihydroxy-(2*R*)-flavanone	*Zygosaccharomyces rouxii* F51 >125.0 μg mL^−1^	*Desmodium caudatum*	[63]
8-(γ,γ-dimethylallyl)-5,7,4′-trihydroxy-(2*R*,3*R*)-dihydroflavonol	*Zygosaccharomyces rouxii* F51 125.0 μg mL^−1^	*Desmodium caudatum*	[63]
8-(γ,γ-dimethylallyl)-5,7,4′-trihydroxy-3′-methoxy-(2*R*,3*R*)-dihydroflavonol	*Zygosaccharomyces rouxii* F51 125.0 μg mL^−1^	*Desmodium caudatum*	[63]
8,3′-di(γ,γ-dimethylallyl)-5,7,4′-trihydroxy-(2*R*,3*R*)-dihydroflavonol	*Zygosaccharomyces rouxii* F51 >125.0 μg mL^−1^	*Desmodium caudatum*	[63]
8-prenylnaringenin	*Staphylococcus aureus* and *Staphylococcus epidermidis,* 12.5 μg mL^−1^	*Glycyrrhiza inflata*	[64]
Abyssinone V-4′ methyl ether	*Bacillus cereus* and *Staphylococcus aureus,* 26–59 μg mL^−1^	*Erythrina lysistemon*	[58]
Alpumisoflavone	*Bacillus cereus*, *Staphylococcus aureus,* and *Pseudomonas aeruginosa,* 20–31 μg mL^−1^	*Erythrina lysistemon*	[58]
Auriculatin	*Staphylococcus aureus*, *Staphylococcus epidermidis,* and *Bacillus subtilis,* 2 μg mL^−1^	*Millettia extensa*	[66]
Cristacarpin	*Pseudomonas aeruginosa* 78 μg mL^−1^	*Erythrina lysistemon*	[58]
Dehydroglyceollidin II	*Staphylococcus aureus* and MRSA, 68 μM	ND ^‡^	[57]
Dehydroglyceollin I	*Staphylococcus aureus* and MRSA, 49 μM	ND ^‡^	[57]
Dehydroglyceollin II	*Staphylococcus aureus* and MRSA, 59 μM	ND ^‡^	[57]
Dehydroglyceollin IV	*Staphylococcus aureus* and MRSA, 130 μM	ND ^‡^	[57]
Dihydrochalcone 1	*Bacillus subtilis* 3.1 μM	*Eriosema montanum*	[56]
Dihydrochalcones 2	*Bacillus subtilis* 7.7 μM	*Eriosema montanum*	[56]
Erybraedin A	*Bacillus cereus*, *Staphylococcus aureus*, *Staphylococcus epidermidis,* and *Escherichia coli,* 1–2 μg mL^−1^	*Erythrina lysistemon*	[58]
Erysubin F	MRSA 15.4 μM	*Erythrina sacleuxii*	[60]
Eryzerin C	*Staphylococcus aureus*, *Staphylococcus epidermidis*, *Escherichia coli,* and *Pseudomonas aeruginosa,* 2–5 μg mL^−1^	*Erythrina lysistemon*	[58]
Flavanonol Ms-II	*Plasmodium falciparum* D6, 3D7, and KSM 009 strains, 1.5–4.6 μM	*Tephrosia subtriflora*	[69]
Fremontone	MRSA 12.5 μg mL^−1^	*Psorothamnus schottii*	[70]
Glabrene	*Staphylococcus aureus* and MRSA, 78 μM	ND ^‡^	[57]
Glabrene	*Staphylococcus aureus* 1199B, 1199, and K1758 strains, 12.5–25 μg mL^−1^	*Glycyrrhiza glabra* *Glycine max*	[71]
Glabridin	*Staphylococcus aureus* and MRSA, 39 μM	ND ^‡^	[57]
Glabrol	*Staphylococcus aureus* and MRSA, 24 μM	ND ^‡^	[57]
Glyceollidin II	*Staphylococcus aureus* and MRSA, 129 μM	ND ^‡^	[57]
Glyceollin I	*Staphylococcus aureus* and MRSA, 296 μM	ND ^‡^	[57]
Glyceollin I	*Staphylococcus aureus* 1199B, 1199, and K1758 strains, 60–80 μg mL^−1^	*Glycyrrhiza glabra* *Glycine max*	[71]
Glyceollin II	*Staphylococcus aureus* and MRSA, 443 μM	ND ^‡^	[57]
Glyceollin III	*Staphylococcus aureus* and MRSA, 296 μM	ND ^‡^	[57]
Glyceollin III	*Staphylococcus aureus* 1199B, 1199, and K1758 strains, >50 μg mL^−1^	*Glycyrrhiza glabra* *Glycine max*	[71]
Glyceollin IV	*Staphylococcus aureus* and MRSA, 123 μM	ND ^‡^	[57]
Glycycoumarin	*Staphylococcus aureus*, *Proteus vulgaris*, *Salmonella typhimurium*, *Escherichia coli*, *Pseudomonas aeruginosa,* and *Candida albicans,* 16–62 μg mL^−1^	*Glycyrrhiza iconica*	[55]
Hispaglabridin A	*Staphylococcus aureus* and MRSA, 111 μM	ND ^‡^	[57]
Iconisoflavan	*Staphylococcus aureus*, *Proteus vulgaris*, *Salmonella typhimurium*, *Escherichia coli*, *Pseudomonas aeruginosa,* and *Candida albicans,* 8–62 μg mL^−1^	*Glycyrrhiza iconica*	[55]
Iconisoflaven	*Staphylococcus aureus*, *Proteus vulgaris*, *Salmonella typhimurium*, *Escherichia coli*, *Pseudomonas aeruginosa,* and *Candida albicans,* 16–125 μg mL^−1^	*Glycyrrhiza iconica*	[55]
Isolicoleafol	*Staphylococcus aureus* and *Staphylococcus epidermidis,* 25 μg mL^−1^	*Glycyrrhiza inflata*	[64]
Isowighteone	*Staphylococcus aureus* and MRSA, 65 μM	ND ^‡^	[57]
Licoisoflavone A	*Staphylococcus aureus* and MRSA, 71 μM	ND ^‡^	[57]
Licoleafol	*Staphylococcus aureus* and *Staphylococcus epidermidis,* 100 MIC > 100 μg mL^−1^	*Glycyrrhiza inflata*	[64]
Licorisoflavan A	*Staphylococcus aureus*, *Proteus vulgaris*, *Salmonella typhimurium*, *Escherichia coli*, *Pseudomonas aeruginosa,* and *Candida albicans,* 2–62 μg mL^−1^	*Glycyrrhiza iconica*	[55]
Lupinalbin A	*Bacillus subtilis* 27 μM	*Eriosema montanum*	[56]
Lupinifolin	*Staphylococcus aureus* 8 μg mL^−1^	*Derris reticulata*	[72]
Luteone	*Staphylococcus aureus* and MRSA, 71 μM	ND ^‡^	[57]
Lysisteisoflavone	*Bacillus cereus* and *Escherichia coli,* 2–6 μg mL^−1^	*Erythrina lysistemon*	[58]
Malheurans A	*Streptococcus mutans*, *Bacillus cereus*, OSSA, and ORSA, 3.3–4.3 μg mL^−1^	*Dalea searlsiae*	[59]
Malheurans B	*Streptococcus mutans*, *Bacillus cereus*, OSSA, and ORSA, 2.7–4.7 μg mL^−1^	*Dalea searlsiae*	[59]
Malheurans C	*Streptococcus mutans*, *Bacillus cereus*, OSSA, and ORSA, 3.0–4.6 μg mL^−1^	*Dalea searlsiae*	[59]
Malheurans D	*Streptococcus mutans*, *Bacillus cereus*, OSSA, and ORSA, 2.0–6.5 μg mL^−1^	*Dalea searlsiae*	[59]
Millexatin A	*Staphylococcus aureus*, *Staphylococcus epidermidis,* and *Bacillus subtilis,* 2 μg mL^−1^	*Millettia extensa*	[66]
Millexatin B	*Staphylococcus epidermidis* and *Staphylococcus aureus,* 128 μg mL^−1^	*Millettia extensa*	[67]
Millexatin D	*Micrococcus luteus*, *Bacillus cereus,* and *Staphylococcus aureus,* 8–32 μg mL^−1^	*Millettia extensa*	[67]
Millexatin F	*Staphylococcus aureus*, *Staphylococcus epidermidis* and *Bacillus subtilis* 2 μg mL^−1^	*Millettia extensa*	[66]
Millexatin G	*Staphylococcus epidermidis* and *Staphylococcus aureus,* 128 μg mL^−1^	*Millettia extensa*	[67]
Millexatin H	*Staphylococcus epidermidis* and *Staphylococcus aureus,* 128 μg mL^−1^	*Millettia extensa*	[67]
Millexatin I	*Micrococcus luteus*, *Streptococcus mutans*, *Staphylococcus epidermidis*, *Bacillus cereus,* and *Staphylococcus aureus,* 64–128 μg mL^−1^	*Millettia extensa*	[67]
Millexatin K	*Micrococcus luteus*, *Streptococcus mutans*, *Staphylococcus epidermidis*, *Bacillus cereus,* and *Staphylococcus aureus,* 32–128 μg mL^−1^	*Millettia extensa*	[67]
Millexatin L	*Micrococcus luteus*, *Streptococcus mutans*, *Staphylococcus epidermidis*, *Bacillus cereus,* and *Staphylococcus aureus,* 32–128 μg mL^−1^	*Millettia extensa*	[67]
Millexatin M	*Staphylococcus epidermidis*, *Bacillus cereus,* and *Staphylococcus aureus,* 128 μg mL^−1^	*Millettia extensa*	[67]
Millipurone	*Micrococcus luteus*, *Streptococcus mutans*, *Staphylococcus epidermidis*, *Bacillus cereus,* and *Staphylococcus aureus,* 2–32 μg mL^−1^	*Millettia extensa*	[67]
Mundulinol	*Plasmodium falciparum* D6, 3D7, and KSM 009 strains, 22.3–35.6 μM	*Tephrosia subtriflora*	[69]
Neobavaisoflavone	*Staphylococcus aureus* and MRSA, 116 μM	ND ^‡^	[57]
Neobavaisoflavone	*Staphylococcus aureus* 1199B, 1199, and K1758 strains, 12.5 μg mL^−1^	ND ^‡^	[71]
Phaseol	*Listeria monocytogenes* and *Escherichia coli,* 69–108 μM	*Glycine max* (L.) Merrill	[68]
Phaseollidin	*Bacillus cereus*, *Staphylococcus aureus*, *Staphylococcus epidermidis*, *Escherichia coli,* and *Pseudomonas aeruginosa,* 5–20 μg mL^−1^	*Erythrina lysistemon*	[58]
Prostratol F	*Streptococcus mutans*, *Bacillus cereus*, OSSA, and ORSA, 5.3–8.0 μg mL^−1^	*Dalea searlsiae*	[59]
Rhodacarpin	*Plasmodium falciparum* 3D7 strain 10.2 ± 0.2 μM	*Tephrosia rhodesica*	[62]
Rhodiflavan A	*Plasmodium falciparum* 3D7 strain 7.3 ± 1.8 μM	*Tephrosia rhodesica*	[62]
Rhodiflavan B	*Plasmodium falciparum* 3D7 strain 5.7 ± 1.9 μM	*Tephrosia rhodesica*	[62]
Rhodiflavan C	*Plasmodium falciparum* 3D7 strain 7.0 ± 2.4 μM	*Tephrosia rhodesica*	[62]
Scandenone	*Staphylococcus aureus*, *Staphylococcus epidermidis,* and *Bacillus subtilis,* 2 μg mL^−1^	*Millettia extensa*	[66]
Sophoraflavanone G	*Staphylococcus aureus* and MRSA, 64 μM	ND ^‡^	[57]
Spinosaflavanone B	*Plasmodium falciparum* D6, 3D7, and KSM 009 strains, 5.5–6.6 μM	*Tephrosia subtriflora*	[69]
Subtriflavanonol	*Plasmodium falciparum* D6, 3D7, and KSM 009 strains, 12.5–24.2 μM	*Tephrosia subtriflora*	[69]
Tachrosin	*Plasmodium falciparum* D6 strain 27.1 ± 3.2 μM	*Tephrosia purpurea* subsp. *leptostachya*	[61]
Terpurlepflavone	*Plasmodium falciparum* D6 strain 14.8 ± 3.2 μM	*Tephrosia purpurea* subsp. *leptostachya*	[61]
Topazolin	*Staphylococcus aureus*, *Proteus vulgaris*, *Salmonella typhimurium*, *Escherichia coli*, *Pseudomonas aeruginosa,* and *Candida albicans,* 2–62 μg mL^−1^	*Glycyrrhiza iconica*	[55]
Vatairenone C	MRSA 29.6 ± 0.8 μM	*Vatairea guianensis*	[54]
Vatairenone D	MRSA and *Enterococcus faecium,* 37.0–80.6 μM	*Vatairea guianensis*	[54]
Vatairenone E	MRSA 49.0 ± 3.4 μM	*Vatairea guianensis*	[54]
Wighteone	*Staphylococcus aureus* and MRSA, 46 μM	ND ^‡^	[57]

* (unless otherwise specified); MIC (minimum inhibitory concentration); ND (not determined); ^‡^ (compounds obtained from plants of the Fabaceae family); IC_50_ (half-maximal inhibitory concentration).

**Table 4 ijms-25-13036-t004:** Anti-inflammatory activity of PFs of the Fabaceae family.

Compound	Bioactivity	Species	References
(2*R*)-3α,7,4′-trihydroxy-5-methoxy-8-(γ,γ-dimethylallyl)-flavanone	Inhibit NF-κB and JNK/AP-1, Down-regulated NO (iNOS), IL-6, TNF-α, and MCP-1.	*Sophora flavescens*	[77]
2-[{2-(1-hydroxy-1-methylethyl)-7-(3-methyl-2-butenyl)-2′,3-dihydrobenzofuran}-5-yl]-7-hydroxy-8-(3-methyl-2-butenyl)chroman-4-one	Reduced IL-1β and IL in LPS and tumor necrosis factor-α-stimulated RAW264.7 cells.	*Sophora tonkinensis*	[86]
2-[{3-hydroxy-2′,2-dimethyl-8-(3-methyl-2-butenyl)}chroman-6-yl]-7-hydroxy-8-(3-methyl-2-butenyl)-chroman-4-one	Reduced IL-1β and IL in LPS and tumor necrosis factor-α-stimulated RAW264.7 cells.	*Sophora tonkinensis*	[86]
3-methoxymundulin	LPS-induced Il1b, Il6, and Ptgs2.	*Genista tridentata*	[83]
5,7-dihydroxy-6-methyl-8-prenyl-4′-methoxy-flavanone	Topical applications.	*Eysenhardtia platycarpa*	[81]
5,7-dihydroxy-6-methyl-8-prenylflavanone	Topical applications.	*Eysenhardtia platycarpa*	[81]
5,7-dihydroxy-6-prenylflavanone	Topical applications.	*Eysenhardtia platycarpa*	[81]
5-hydroxy-7-methoxy-6-prenylflavanone	Topical applications.	*Eysenhardtia platycarpa*	[81]
5-MethyIsophoraflavanone B	COX-2 inhibition: IC_50_ > 100 μM.	*Sophora flavenscens*	[75]
8-prenyl daidzein	Repress NF-κB activation, which reduce ERK1/2, JNK, and p38 MAPK.	*Glycine max*	[87]
8-prenyl genistein	Repress NF-κB activation, which reduce ERK1/2, JNK, and p38 MAPK.	*Glycine max*	[87]
Bavachromene	Inhibition of NO production in LPS-activated microglia. IC_50_ = 2.4 ± 0.18 μM.	*Cullen corylifolium*	[84]
Echinoisoflavanone	Down-regulated COX-2 expression at 10–25 μM.	*Sophora fflvenscens*	[75]
Echinoisosophoranone	COX-2 inhibition: IC_50_ > 100 μM.	*Sophora flavenscens*	[75]
Hedysarumine B	Inhibition of NO production in BV-2 cells. IC_50_ = 5.12 ± 0.19 μM.	*Hedysarum gmelinii*	[42]
Hedysarumine E	Inhibition of NO production in BV-2 cells. IC_50_ = 12.41 ± 2.07 μM.	*Hedysarum gmelinii*	[42]
Hedysarumine G	Inhibition of NO production in BV-2 cells. IC_50_ = 3.25 ± 0.20 μM.	*Hedysarum gmelinii*	[42]
Isobavachalcone	Inhibition of NO production in LPS-activated microglia. IC_50_ = 1.6 ± 0.11 μM.	*Cullen corylifolium*	[84]
Isobavachin	Inhibit MAPK and NF-κB.	*Psoralea corylifolia*	[80]
Isosophoranone	COX-2 inhibition: IC_50_ > 100 μM.	*Sophora flavenscens*	[75]
Kanzonol B	Inhibition of NO production in LPS-activated microglia. IC_50_ 2.2 ± 0.21 μM.	*Cullen corylifolium*	[84]
Kuraridin	Down-regulated COX-2 expression at 10–25 μM.	*Sophora flavenscens*	[75]
Kurarinone	Down-regulated COX-2 expression at 10–25 μM.	*Sophora flavenscens*	[75,76]
Kushenol C	STAT1, STAT6 and NF-κB. IC_50_ = 50–100 μM.	*Sophora flavescens*	[79]
Leachianone A	Inhibits interleukin (IL)-6, IL-8, and CXCL1.	*Sophora flavescens*	[85]
Lupinifolin	LPS-induced Il1b, Il6, and Ptgs2.	*Genista tridentata*	[83]
Mundulin	LPS-induced Il1b, Il6, and Ptgs2.	*Genista tridentata*	[83]
Paratocarpin A	Inhibition of NO production in BV-2 cells. IC_50_ = >100 μM.	*Hedysarum gmelinii*	[42]
Paratocarpin B	Inhibition of NO production in BV-2 cells. IC_50_ = 18.18 ± 1.27 μM.	*Hedysarum gmelinii*	[42]
Paratocarpin E	Inhibition of NO production in BV-2 cells. IC_50_ = 10.33 ± 0.26 μM.	*Hedysarum gmelinii*	[42]
Paratocarpin F	Inhibition of NO production in BV-2 cells. IC_50_ = 8.48 ± 2.52 μM.	*Hedysarum gmelinii*	[42]
Sanggenon B	COX-2 inhibition: IC_50_ > 100 μM.	*Sophora flavenscens*	[75]
Sanggenon D	Down-regulated COX-2 expression at 10–25 μM.	*Sophora flavenscens*	[75]
Sophoraflavanone G	Down-regulated COX-2 expression at 10–25 μM.	*Sophora flavenscens*	[75]
Sophoraflavanone G	PI3K/Akt, JAK/STAT, and Nrf2/HO-1 Pathways. IC_50_ = 10–30 μM.	*Sophora alopecuroides*	[78]
Sophoraflavanone G	Inhibits interleukin (IL)-6, IL-8, and CXCL1.	*Sophora flavescens*	[85]

IC_50_ (half-maximal inhibitory concentration).

**Table 5 ijms-25-13036-t005:** Anti-diabetic activity of PFs found in plants from the Fabaceae family.

Compounds	Target (IC_50_)	Species	Reference
4′,1″-dihydroxy-3′-methoxy-6,7-furanflavanone	*α*-glucosidase inhibition: IC_50_ = 53.1 μg mL^−1^	*Psoralea corylifolia*	[91]
4′*-O*-methylbavachalcone	PPAR-*γ* agonist activity = 25 μM	*Psoralea corylifolia*	[94]
5,3′,4′-trihydroxy-1″-methoxy-6,7-furanbavachalcone	*α*-glucosidase inhibition: IC_50_ = 90.3 μg mL^−1^	*Psoralea corylifolia*	[91]
5′-prenyleriodictyol	*α*-glucosidase inhibition: IC_50_ = 53.4 μg mL^−1^	*Glycyrrhiza uralensis*	[88]
6-prenyleriodictyol	*α*-glucosidase inhibition: IC_50_ = 27.5 μg mL^−1^	*Glycyrrhiza uralensis*	[88]
6-prenylnaringenin	*α*-glucosidase inhibition: IC_50_ = 15.4 μg mL^−1^	*Glycyrrhiza uralensis*	[88]
6-Prenylquercetin	*α*-glucosidase inhibition: IC_50_ = 3.7 μg mL^−1^	*Glycyrrhiza uralensis*	[88]
Bavachin	PPAR-*γ* agonist activity = 25 μM	*Psoralea corylifolia*	[94]
Bavachinin	PPAR-*γ* agonist activity = 25 μM	*Psoralea corylifolia*	[94]
Broussochalcone B	PPAR-*γ* agonist activity = 25 μM	*Psoralea corylifolia*	[94]
Corylifol A	PPAR-*γ* agonist activity = 25 μM	*Psoralea corylifolia*	[94]
Dioxycudraflavone A	*α*-glucosidase inhibition: IC_50_ = 25.27 μg mL^−1^	*Glycyrrhiza uralensis*	[88]
Hirtacoumaroflavonoside	*α*-glucosidase inhibition: IC_50_ = 22 μM	*Glycyrrhiza uralensis*	[89]
Hirtaflavonoside B	*α*-glucosidase inhibition: IC_50_ = 71 μM	*Glycyrrhiza uralensis*	[89]
Isobavachalcone	PPAR-*γ* agonist = 18 μM	*Psoralea corylifolia*	[94]
Isobavachin	PPAR-*γ* agonist = 22 μM	*Psoralea corylifolia*	[94]
Morusinol	*α*-glucosidase inhibition: IC_50_ = 23.2 μg mL^−1^	*Glycyrrhiza uralensis*	[88]

IC_50_ (half-maximal inhibitory concentration).

**Table 6 ijms-25-13036-t006:** Anti-Alzheimer’s and neuroprotective activities of PFs of the Fabaceae family.

Compound	IC_50_ (μM)	Species	Reference
(−)-(2*S*)-2′,4′-dihydroxy-5′-(1‴,1‴-dimethylallyl)-8-prenylpinocembrin (8 PP)	12.5 ± 3.9	*Dalea elegans*	[95]
(−)-(2*S*)-3′,4′-dihydroxy-6,2′-diprenylpinocembrin (pazentin A)	>300	*Dalea pazensis*	[95]
(−)-(2*S*)-4′-hydroxy-2′-methoxy-5′-(1‴,1‴-dimethylallyl)-8-prenylpinocembrin (Me8PP)	>300	*Dalea elegans*	[95]
(−)-(2*S*)-8-prenylpinocembrin (glabranin)	44.5 ± 1.2	*Dalea elegans*	[95]
(2*S*)-(−)-4′-hydroxy-2′-methoxy-5′-(1‴,1‴-dimethylallyl)-6-prenylpinocembrin (pazentin B)	40.7 ± 0.6	*Dalea pazensis*	[95]
Bavachalcone	6.10	*Psoralea fructus*	[105]
Bavachin	7.71	*Psoralea fructus*	[105]
Bavachinin	27.06	*Psoralea fructus*	[105]
Isobavachalcone	19.32	*Psoralea ructus*	[105]

IC_50_ (half-maximal inhibitory concentration).

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
