# Peer review of "Therapeutic Potential of Prenylated Flavonoids of the Fabaceae Family in Medicinal Chemistry: An Updated Review"

_ijms, 2024, doi:10.3390/ijms252313036_

Round 1
Reviewer 1 Report (Previous Reviewer 1)
Comments and Suggestions for Authors
1. In “Introduction”, background introduction seemed too long and not clear. Too much about “Polyphenols” and “Flavonoids”. In this study, the focus was “Prenylated flavonoids” from Fabaceae, which should be stated clearly at first.
2. Figure 2 was not well designed, organized and drawn. Some structures were big and some were small. They were not of a uniform configuration.
3. Line 75-80, 260 PFs were classified into 5 groups, anti-cancer activity (60 PFs), antimicrobial (128 PFs), anti-inflammatory (36 PFs), anti-diabetic (17 PFs) and anti-Alzheimer’s and neuroprotective activities (9 PFs). 60+128+36+17+9=260?
4. Line 75-80, according to the categories, for an individual PF, it just had one activity? Especially for those PFs with anti-cancer activity and anti-Alzheimer’s and neuroprotective activity, they did not show anti-inflammatory activity? In my opinion, this classification method was not right.
5. Line 126-127, “Most of the articles collected in these two databases focused on the Fabaceae family (Figure 3a)”. Figure 3a was about articles on the plants of Fabaceae or PFs from Fabaceae? It was not stated clearly.
6. I did not understand Figure 3b. What did it express? For anti-cancer activity, there were 9 plants, why to show so many sections (more than 9) in the circle?
7. Almost all the pharmacological trials were in vitro. No related papers on in vivo studies were published?
8. “Conclusions” were still not qualified. Nothing valuable from the authors themselves were shown.
9. Though this paper has been modified, it was still not like a review. In addition, many issues on writing logic and scientificity still existed. Above were a part of all the issues.
Comments on the Quality of English LanguageEnglish writing should be improved more.
Author Response
Dear Reviewer, thank you very much for taking the time to review this manuscript. Please find the detailed responses below and the corresponding revisions/additional corrections highlighted in yellow in the re-submitted attached file. We hope that this new corrected version of the manuscript meets your requirements to be finally accepted.
Quality of English Language
The English could be improved to more clearly express the research.
Response: Thank you for pointing this out. We agree with this comment. Therefore, we have revised and improved the English language to express clearly the investigation. Besides, the final version of the manuscript has been reviewed and corrected by a colleague (native of the USA), who knows the subject of the manuscript.
Point-by-point Responses to Comments and Suggestions:
1. In “Introduction”, background introduction seemed too long and not clear. Too much about “Polyphenols” and “Flavonoids”. In this study, the focus was “Prenylated flavonoids” from Fabaceae, which should be stated clearly at first.
Response: Thank you for pointing this out. We agree with this comment. Therefore, we have restructured the introduction, eliminating the first part and adding clear information focused on the prenylated flavonoid compounds of the Fabaceae family (Line: 34-80).
2. Figure 2 was not well designed, organized and drawn. Some structures were big and some were small. They were not of a uniform configuration.
Response: Thank you for pointing this out. We agree with this comment. Therefore, we have reviewed, corrected, add new information, and redrawn Figure 2, maintaining a uniform configuration of the chemical structures throughout the metabolic pathway (Line: 97-115)
3. Line 75-80, 260 PFs were classified into 5 groups, anti-cancer activity (60 PFs), antimicrobial (128 PFs), anti-inflammatory (36 PFs), anti-diabetic (17 PFs) and anti-Alzheimer’s and neuroprotective activities (9 PFs). 60+128+36+17+9=260?
Response: Thank you for pointing this out. We agree with this comment. We apologize for this error; we have introduced 60 instead of 70 compounds with anti-cancer activity, which coincides with 260 PF compounds obtained from several species of the Fabaceae family. Besides, we have modified this part of the text to highlight compounds that showed more than one biological activity (Line: 70-80).
4. Line 75-80, according to the categories, for an individual PF, it just had one activity? Especially for those PFs with anti-cancer activity and anti-Alzheimer’s and neuroprotective activity, they did not show anti-inflammatory activity? In my opinion, this classification method was not right (Line 78-80).
Response: Thank you for pointing this out. We agree with this comment. Therefore, we have identified 11 prenylated flavonoid compounds with more than one biological activity. Furthermore, we have designed Table 1 placing each compound in alphabetical order, marking its biological activity with a check.
5. Line 126-127, “Most of the articles collected in these two databases focused on the Fabaceae family (Figure 3a)”. Figure 3a was about articles on the plants of Fabaceae or PFs from Fabaceae? It was not stated clearly.
Response: Thank you for pointing this out. We agree with this comment. As shown in the title of Figure 3a, most of the articles collected (PubMed and Web of Knowledge) focused on the PF compounds of the Fabaceae family. Based on your valuable comment, we have added clear information from Figure 3a (Line: 126-127).
6. I did not understand Figure 3b. What did it express? For anti-cancer activity, there were 9 plants, why to show so many sections (more than 9) in the circle?
Response: We apologize for this error. During the design of the Figure 3b, some species were accidentally cut. We have corrected the image and added specific and precise information. Each section (in color) of the circle shows the number of compounds identified per Fabaceae species and biological activities: e.g., 15 compounds correspond to the species Glycyrrhiza uralensis, 12 compounds for Desmodium renifolium, 8 compounds for Mappianthus iodoides, etc., for anti-cancer activity. For other activities, a similar procedure was followed (Line: 132-135).
7. Almost all the pharmacological trials were in vitro. No related papers on in vivo studies were published?
Response: Thank you for pointing this out. We agree with this comment. Therefore, we have added specific and precise information from in vivo testing of PF compounds isolated from Fabaceae plants, although previous investigations on plant PF bioactivities in animal models are mostly limited (Line: 237-245; 407-411; 584-596).
8. “Conclusions” were still not qualified. Nothing valuable from the authors themselves were shown.
Response: Thank you for pointing this out. We agree with this comment. Therefore, we have qualified the conclusions, highlighting the therapeutic potential of the prenylated flavonoid compounds of the Facaceae family, highlighting those with more significant biological activity, such as isabavachalcone and soforaflavanone G as excellent candidates for testing in animal models, due to their high biological activities: anti-inflammatory, antidiabetic, anticancer and anti-Alzheimer and neuroprotective activity, respectively (Line: 668-679).
9. Though this paper has been modified, it was still not like a review. In addition, many issues on writing logic and scientificity still existed. Above were a part of all the issues.
Response: Thank you for pointing this out. We agree with this comment. Therefore, we have reviewed each part of the manuscript, considering the logic and scientificity of this new version. Besides, the final version of the manuscript has been reviewed and corrected by a colleague (native of the USA), who knows the subject of the manuscript.
Comments on the Quality of English Language
English writing should be improved more.
Response: Thank you for pointing this out. We agree with this comment. Therefore, we have revised and improved the English writing to express the investigation clearly. Furthermore, Besides, the final version of the manuscript has been reviewed and corrected by a colleague (native of the USA), who knows the subject of the manuscript.
Reviewer 2 Report (New Reviewer)
Comments and Suggestions for Authors
Reviewer comments
The review article entitled “Therapeutic Potential of Prenylated Flavonoids of the Fabaceae Family in Medicinal Chemistry: An Updated Review” by Jaime Morante-Carriel et al. is very appealing and the idea is attractive. It was written carefully in good and understandable English. This paper fully reaches the scientific level to be published at the International Journal of Molecular Sciences, and I recommend it for this journal after few typos points to adress.
Here are few comments to be addressed:
̶ Line 133 “Cancer, a leading causes of death” change “causes” to “cause”.
̶ Line 143 “ ….new PFs with different biological activity against different types …” change “activity” to “activities”
̶ Line 152: Make “in vivo” italic
̶ An apostrophe “’s” isn’t used for objects. So please remove all apostrophe ’s in the words: “auriculasin's” line 210 to be “auriculasin B-ring”, “microglia's nitric oxide” Line 440 to be “microglia nitric oxide”, “world’s population” Line 586 to be “world population”.
̶ Line 266: write the names of the two new prenylated dihydrochalcones.
̶ Line 351: make the 50 in IC50 subscript.
̶ Line 416: “methosy-6-prenylflavanone …..” change “methosy” to “methoxy”
̶ Line 532: there is a space between the 4′ and the -O in 4′-O-methylbavachalcone. Please remove.
̶ In table 4: in the second row there is an extra digit one “1.” on the left. Please remove
̶ At the beginning of Line 570 “In G. max” isn’t understandable, please explain the abbreviation or remove.
Author Response
Dear Reviewer, thank you very much for taking the time to review this manuscript. Please find the detailed responses below and the corresponding revisions/additional corrections highlighted in Turquoise in the attached re-submitted file. We hope that this new corrected version of the manuscript meets your requirements to be finally accepted.
Point by point Responses to Comments and Suggestions:
̶ Comment 1: Line 133 “Cancer, a leading causes of death” change “causes” to “cause”.
Response: Thank you for pointing this out. We agree with this comment. Therefore, we have changed the word "causes" to "cause" (Line 137).
̶ Comment 2: Line 143 “ …new PFs with different biological activity against different types …” change “activity” to “activities”
Response: Thank you for pointing this out. We agree with this comment. Therefore, we have changed the word "activity" to "activities" (Line 147).
̶ Comment 3: Line 152: Make “in vivo” italic
Response: Thank you for pointing this out. We agree with this comment. Therefore, we have placed the word "in vitro" in italics (Line 156).
̶ Comment 4: An apostrophe “’s” isn’t used for objects. So please remove all apostrophe ’s in the words: “auriculasin's” line 210 to be “auriculasin B-ring”, “microglia's nitric oxide” Line 440 to be “microglia nitric oxide”, “world’s population” Line 586 to be “world population”.
Response: Thank you for pointing this out. We agree with this comment. Therefore, we have removed the apostrophe in words: “auriculasin's” to be “auriculasin B-ring (line 214); “microglia's to be “microglia nitric oxide” (line 454); “world’s population” to be “world population” (line 613).
̶ Comment 5: Line 266: write the names of the two new prenylated dihydrochalcones.
Response: Thank you for pointing this out. We agree with this comment. Therefore, we have written the names of the two new prenylated dihydrochalcones ((2′,4′,5,6′-tetrahydroxy-4-methoxy-3,3′-diprenyldihydrochalcone,2′,4′,4,6′-tetrahydroxy-3,3′-diprenyldihydrochalcone and lupinalbin A), modifying the wording slightly (Line: 282-283).
̶ Comment 6: Line 351: make the 50 in IC50 subscript.
Response: Thank you for pointing this out. We agree with this comment. Therefore, we have subscripted the 50 in IC50 throughout the manuscript (Line 162, and throughout the manuscript).
̶ Comment 7: Line 416: “methosy-6-prenylflavanone ….” change “methosy” to “methoxy”
Response: Thank you for pointing this out. We agree with this comment. Therefore, we have changed “methosy” to “methoxy” to be methoxy-6-prenylflavanone (Line 430).
̶ Comment 8: Line 532: there is a space between the 4′ and the -O in 4′-O-methylbavachalcone. Please remove.
Response: Thank you for pointing this out. We agree with this comment. Therefore, we have eliminated the space between the 4′ and the -O in 4′-O-methylbavachalcone (Line 546).
̶ Comment 9: In table 4: in the second row there is an extra digit one “1.” on the left. Please remove
Response: Thank you for pointing this out. We agree with this comment. Therefore, we have eliminated the number "1" placed incorrectly on the left side of 4′-O-methylbavachalcone compound in the second row of Table 4.
̶ Comment 10: At the beginning of Line 570 “In G. max” isn’t understandable, please explain the abbreviation or remove.
Response: Thank you for pointing this out. We agree with this comment. Therefore, we have corrected the abbreviation G. max, improving the wording of the text for better understanding (Lines 597-598).
Comments on the Quality of English Language
We have revised and improved the English writing to express clearly the investigation. The final version of the manuscript has been reviewed and corrected by a colleague (native of the USA), who knows the subject of the manuscript.
This manuscript is a resubmission of an earlier submission. The following is a list of the peer review reports and author responses from that submission.
Round 1
Reviewer 1 Report
Comments and Suggestions for Authors
1. English writing should be improved by a professional editing service, not only for grammar but also for writing logic and reading fluency.
(1) Line 26, “critically” seemed more suitable than “critical”.
(2) Line 29-30, “pharmacological activities” seemed more suitable than “pharmacological activity”.
(3) Between the two sentences of Line 62-64, it is better to list some plants belonging to Fabaceae.
(4) In Figure 1, the chemical stucrture of licoisoflavone A was not right.
2. All the prenylated flavonoids in Fabaceae plants were shown in Figure 1? In my opinion, this category of components in Fabaceae should be demonstrated as much as possible. This paper was focused on “Therapeutic Potential of Prenylated Flavonoids of the Fabaceae Family”. At least, the authors should list all these prenylated flavonoids and their original plants, by which the literature concerning the therapeutic potential can be stated completely and clearly.
3. “2. Biosynthetic pathway of prenylated flavonoids” seemed too simple and nothing valuable was mentioned. In my opinion, the pathways can be drawn in a figure and be stated with some words.
4. In Figure 2b, for each activity, the number of PFs in a plant was shown. Were they repetitive? For example, for “3.5. Anti-Alzheimer and neuroprotective activity”, was there a PF repeated in 3, 3 and 4 of the three plants?
5. “3. Biological activities of prenylated flavonoids”,
(1) Tables showed almost same contents as text. I think it was not necessary.
(2) Almost all the activities were in vitro. Were there published papers of in vivo studies? If so, why not to cite them?
(3) Nothing valuable was shown in this part. Just PF names, IC50 or some concentrations, cell names and plant names were listed without any in-depth mechanisms.
6. Why not to analyze the relationships between the structure of PF and the activities?
7. After reading this paper, I did not understand the aim or purpose of writing it. Just demonstration?
8. This paper is a review. However, existing issues, research orientations and future perspective were not mentioned. Just a simple “4. Conclusions” was shown. This paper is more like a “sum” of published literature. Nothing of the authors themselves were demonstrated. So, it is not qualified as a review.
Comments on the Quality of English LanguageEnglish writing should be improved by a professional editing service, not only for grammar but also for writing logic and reading fluency.
Author Response
Dear Reviewer,
Thank you very much for your Comments and Suggestions:
Suggestions: English writing should be improved by a professional editing service, not only for grammar but also for writing logic and reading fluency.
Response: We agree with this suggestion. English grammar and writing has been improved by a native, expert on the topic discussed.
Main remarks:
Corrections are highlighted in yellow in the attached manuscript
Comments 1: Line 26, “critically” seemed more suitable than “critical”.
Response: Thank you for pointing this out. We agree with this comment. Therefore, we have
replaced “critical” with “critically” in line 26.
Comments 2: Line 29-30, “pharmacological activities” seemed more suitable than “pharmacological activity”.
Response: Thank you for pointing this out. We agree with this comment. Therefore, we have
replaced the words “pharmacological activity” with “pharmacological activities” in lines 29-30.
Comments 3: Between the two sentences of Line 62-64, it is better to list some plants belonging to Fabaceae.
Response: Thank you for pointing this out. We agree with this comment. Therefore, we have listed some plants belonging to the Fabaceae family between lines 62-64.
Comments 4: In Figure 1, the chemical stucrture of licoisoflavone A was not right.
Response: Thank you for pointing this out. We agree with this comment. Therefore, we have corrected the chemical structure of Licoisoflavone A in Figure 1.
Comments 5: All the prenylated flavonoids in Fabaceae plants were shown in Figure 1? In my opinion, this category of components in Fabaceae should be demonstrated as much as possible. This paper was focused on “Therapeutic Potential of Prenylated Flavonoids of the Fabaceae Family”. At least, the authors should list all these prenylated flavonoids and their original plants, by which the literature concerning the therapeutic potential can be stated completely and clearly.
Response: Thank you for pointing this out. We agree with this comment. Listing the compounds may not be beneficial since it would be nothing more than a number in the text. Although it gives a certain fluidity to the reading of the text, it forces the reader to go to the tables to find out what compound it is about; the way in which the manuscript is laid out is self-contained. We apologize for any inconvenience caused by this disagreement.
Comments 6: “2. Biosynthetic pathway of prenylated flavonoids” seemed too simple and nothing valuable was mentioned. In my opinion, the pathways can be drawn in a figure and be stated with some words.
Response: Thank you for pointing this out. We agree with this comment. Therefore, we have included more information in this section, referring to the metabolic pathway of prenylated flavonoid compounds from plants, recently published by our research group in the journal Plants (https://doi.org/10.3390/plants13091211). We consider that including the metabolic pathway of PF compounds could result in repetitive information.
Comments 7: In Figure 2b, for each activity, the number of PFs in a plant was shown. Were they repetitive? For example, for “3.5. Anti-Alzheimer and neuroprotective activity”, was there a PF repeated in 3, 3 and 4 of the three plants?.
Response: Thank you for pointing this out. We agree with this comment. Therefore, the number of FPs were not repetitive in each activity. Some plants of the Fabaceae family are infrequently repeated in some bioactivities discussed in our manuscript.
Comments 8: “3. Biological activities of prenylated flavonoids”: (1) Tables showed almost same contents as text. I think it was not necessary.
Response: Thank you for pointing this out. We agree with this comment. Therefore, we have
reviewed and corrected the tables, avoiding repetition with the text.
Comments 9: (2) Almost all the activities were in vitro. Were there published papers of in vivo studies? If so, why not to quote them?
Response: Thank you for pointing this out. We agree with this comment. Although in vivo studies in Fabaceae family are limited, we have included some paragraphs from these studies in our manuscript.
Comments 10: (3) Nothing valuable was shown in this part. Just PF names, IC50 or some concentrations, cell names and plant names were listed without any in-depth mechanisms.
Response: We have made important adjustments to the manuscript, specifically highlighting the name of the compounds, plant species, biological activity, and authorship. We have also highlighted the compounds with the greatest activity for each case in relation to those most cited in the literature.
Comments 11: (3) 6. Why not to analyze the relationships between the structure of PF and the activities?
Response: Thank you for pointing this out. We agree with this comment. Although our work does not particularly focus on the chemical structures of PFs, we have included some structures of the compounds that show greater anti-cancer, anti-diabetic, antimicrobial, anti-inflammatory, anti-Alzheimer’s and neuroprotective activities, relating structure to biological activities.
Comments 12: 7. After reading this paper, I did not understand the aim or purpose of writing it. Just demonstration?
Response: Thank you for pointing this out. We agree with this comment. Although the objective of our manuscript is to present the latest scientific findings on the biological activities of PF compounds from Fabaceae plants and the associated health benefits supported by experimental evidence from the last decade, we have restructured some paragraphs and included new relevant information to clarify the purpose, identifying existing problems, research directions, and future perspectives.
Comments 13: 8. This paper is a review. However, existing issues, research orientations and future perspective were not mentioned. Just a simple “4. Conclusions” was shown. This paper is more like a “sum” of published literature. Nothing of the authors themselves were demonstrated. So, it is not qualified as a review.
Response: Thank you for pointing this out. We agree with this comment. However, in relation to the previous comment, the new information included in the manuscript has allowed us to make a solid conclusion, demonstrating the therapeutic potential of PFs of the Fabaceae family in medicinal chemistry.
Comments on the Quality of English Language:
Comment: English writing should be improved by a professional editing service, not only for grammar but also for writing logic and reading fluency.
Response: We agree with this comment. English grammar and logical writing has been improved by a native, with experience on the topic discussed.
Reviewer 2 Report
Comments and Suggestions for Authors
Review of the article "Therapeutic Potential of Prenylated Flavonoids of the Fabaceae Family in Medicinal Chemistry. An Updated Review" by Morante-Carriel et al.
The authors describe the potential therapeutic properties of prenylated flavonoids from the Fabaceae family. It is worth noting that the presented information concerns mostly the last 5 years (when looking at references). The introduction briefly but factually introduces the topic of the review. Then the authors briefly describe the biosynthetic pathways of prenylated flavonoids, referring the reader for detailed information to the source literature. The main part consists of information on the biological properties of various prenylated flavonoids: anti-cancer, antimicrobial, anti-inflammatory and anti-Alzheimer activity. It is a pity that the authors do not present data on antioxidant properties.
Main remarks:
1) Throughout the article, Latin names should be written in italics (e.g. line 127, etc.).
2) Throughout the article, "O" in compound names should be italicized, as should "S" and "R" (e.g. line 122, 211, etc., in tables as well)
3) I would suggest adding in the Supplementary Materials all compounds discussed in the manuscript.
4) Table 3. - there are unnecessary ordinal numbers next to compound names, e.g. 1., 2., etc.
5) In vitro and in vivo should be written in italics throughout the manuscript.
Comments on the Quality of English LanguageMinor editing of English language required.
Author Response
Dear Reviewer,
Thank you very much for your Comments and Suggestions:
Responde: In relation with the last part of your comment “It is a pity that the authors do not present data on antioxidant properties”, we apologize for not including a section on antioxidant activity of the Fabaceae family. Currently, we are preparing a new manuscript related to the antioxidant properties of the botanical families available in our research group, including the Fabaceae family.
Main remarks:
Corrections are highlighted in turquoise in the attached manuscript
Comments 1: Throughout the article, Latin names should be written in italics (e.g. line 127, etc.).
Response: Thank you for pointing this out. We agree with this comment. Therefore, we have written the Latin name in italics in line 127 and throughout the manuscript.
Comments 2: Throughout the article, "O" in compound names should be italicized, as should "S" and "R" (e.g. line 122, 211, etc., in tables as well)
Response: Thank you for pointing this out. We agree with this comment. Therefore, we have written "O”, “S”, “R" of the compounds name in italics in lines 122, 211, etc., and also on the tables.
Comments 3: I would suggest adding in the Supplementary Materials all compounds discussed in the manuscript.
Response: Thank you for bringing this to our attention. However, we respectfully disagree with the comment because all the compounds discussed in this review are a fundamental part of the manuscript's main text. If we were to move this information to supplementary material, it could potentially confuse the reader in understanding the connection between the text and the name/structure of prenylated flavonoid compounds. Additionally, in our review of other current literature, we found that none of the discussed compounds were deposited as supplementary material. We apologize for any inconvenience caused by this disagreement.
Comments 4: Table 3. - there are unnecessary ordinal numbers next to compound names, e.g. 1., 2., etc.
Response: Thank you for pointing this out. We agree with this comment. Therefore, we have delete the unnecessary ordinal numbers of compounds names in table 3.
Comments 5: In vitro and in vivo should be written in italics throughout the manuscript.
Response: Thank you for pointing this out. We agree with this comment. Therefore, we have written the in vitro and in vivo terms in italics throughout the manuscript.
Comments on the Quality of English Language:
Comments: Minor editing of English language required.
Response: We agree with this comment. We have reviewed and corrected English grammar throughout the manuscript.
Round 2
Reviewer 1 Report
Comments and Suggestions for Authors
Though the authors have made some revisions, they did not answer all the questions according to the comments and it was not good enough for acceptance on design, organization and writing.
Comments on the Quality of English LanguageEnglish writing was not good enough.